# Automated neuronal reconstruction with super-multicolour Tetbow labelling and threshold-based clustering of colour hues

Marcus N. Leiwe[1,2,3], Satoshi Fujimoto [1,3], Toshikazu Baba[1,3], Daichi Moriyasu [1], Biswanath Saha [1], Richi Sakaguchi [1], Shigenori Inagaki[1] & Takeshi Imai [1] ✉

Fluorescence imaging is widely used for the mesoscopic mapping of neuronal connectivity. However, neurite reconstruction is challenging, especially when neurons are densely labelled. Here, we report a strategy for the fully automated reconstruction of densely labelled neuronal circuits. Firstly, we establish stochastic super-multicolour labelling with up to seven different fluorescent proteins using the Tetbow method. With this method, each neuron is labelled with a unique combination of fluorescent proteins, which are then imaged and separated by linear unmixing. We also establish an automated neurite reconstruction pipeline based on the quantitative analysis of multiple dyes (QDyeFinder), which identifies neurite fragments with similar colour combinations. To classify colour combinations, we develop unsupervised clustering algorithm, dCrawler, in which data points in multi-dimensional space are clustered based on a given threshold distance. Our strategy allows the reconstruction of neurites for up to hundreds of neurons at the millimetre scale without using their physical continuity.

The brain is made up of dense networks of interconnected neurons. Mapping the anatomy of these dense networks is one of the biggest challenges in neuroscience. Electron microscopy (EM) provides the synaptic resolution and is used as a gold standard in connectomics[1,2]. It is now possible to obtain EM images for 1 mm³ volumes (~petabyte scale)[3,4]; however, due to its extremely large data size, the reconstruction process is the bottle neck. Light microscopy (LM) is useful for the mesoscopic circuit mapping at a whole-brain level[5–9]. However, the reconstruction of densely labelled circuits is challenging as its limited resolution hinders the discrimination of thin axonal fibres (down to ~100 nm) originating from different neurons. It is, therefore, essential to limit the number of labelled neurons in single-cell reconstruction in LM. Moreover, manual circuit tracing is a highly laborious and rate-limiting step in large-scale circuit reconstruction with LM. For both EM and LM connectomics, current reconstruction strategies are all based on the continuity of the target structure; we, therefore, cannot reconstruct the neurites correctly once we lose just a few sections of the images. In theory, error rates in reconstruction exponentially increase as the length of the neurites increases. Thus, connectomics beyond a millimetre scale remains a big challenge.

To improve the discriminability of neurites in a densely labelled circuits, it is effective to utilize the colour information in LM. For example, if we utilize the combination of three colours (red, green, and blue), 13 different lines can be easily dissociable in a Tokyo subway map (Fig. 1a). Similarly, multicolour labelling is useful for LM-based circuit reconstruction in the brain. Stochastic multicolour labelling strategies, such as Brainbow, utilize a combination of 3 fluorescent proteins (XFPs) to create different colour hues[10,11]. A brighter version of this method, Tetbow[12], allows for multicolour fluorescence imaging in 3D in combination with tissue clearing with SeeDB2[13]. However, the combination of three colours only produces ~20 discernible colour hues[12], which falls well short of the variations necessary to reconstruct densely labelled neuronal circuits.

[1]Graduate School of Medical Sciences, Kyushu University, Fukuoka, Japan. [2]Present address: MetaCell LCC, LTD, Cambridge, MA, USA. [3]These authors contributed equally: Marcus N. Leiwe, Satoshi Fujimoto, Toshikazu Baba. ✉e-mail: imai.takeshi.457@m.kyushu-u.ac.jp

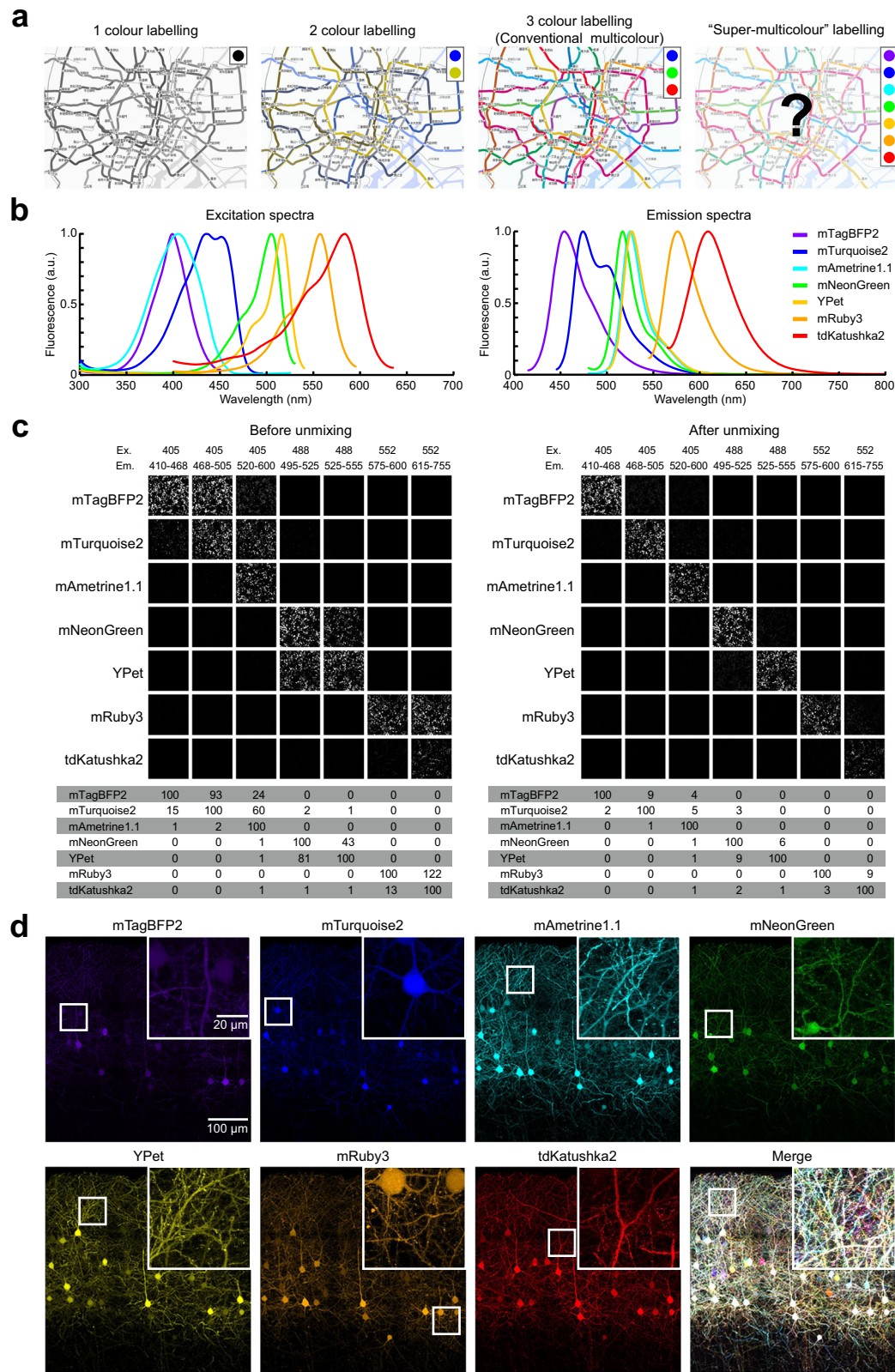

In this study, we demonstrate "super-multicolour" labelling of neuronal circuits, in which >3 XFPs are utilized to expand colour combinations. However, we cannot recognize the combination of >3 primary colours visually, as most humans only have trichromatic colour vision. We, therefore, develop a fully automated neurite reconstruction pipeline based on quantitative analysis of multiple dyes (QDyeFinder). In this pipeline, we identify neurites for different neurons based on colour information only. The combination of super-multicolour labelling and the QDyeFinder pipeline facilitates automated circuit reconstruction beyond a millimetre scale without tedious manual tracing.

**Fig. 1 | 7-colour fluorescence imaging with linear unmixing. a** Cartoons illustrating the concept of super-multicolour fluorescence imaging. Tokyo metro maps shown by the combination of 1-3 primary colours. Multicolour labelling facilitates the identification of different lines. However, using >3 primary colours is beyond the human visual range. Image was modified under CC-BY-SA-3.0 from: https://commons.wikimedia.org/wiki/File:Tokyo_metro_map.png. **b** Excitation and emission spectra of our chosen XFPs for 7-colour Tetbow, highlighting the overlap of emission signals between XFPs. A.U., arbitrary unit. **c** 7 XFPs transfected in HEK293T cells before (left panels) and after (right panels) linear unmixing. Reference data for the linear unmixing was acquired from another set of images (Supplementary Fig. 2a). Percentages of normalized intensities are shown below. See also Supplementary Figs. 2b–d, 3, and 4. **d** Layer 2/3 neurons in S1 were labelled with 7-colour Tetbow using in utero electroporation. Age, postnatal day (P) 28. XFP images after linear unmixing are shown. Z-stacked images of $514.57 \times 513.49 \times 43.48695 \ \mu m^3$. Inset displays magnified images (white box). Source data are provided as a Source Data file.

## Results

### Super-multicolour imaging with 7 XFPs and linear unmixing

Stochastic multicolour labelling of neurons (e.g., Brainbow) typically utilizes only 3 XFPs[11]. This is simply because it is difficult to identify >3 primary colours in the merged images visually. However, the possible combinations or "colour hues" will increase by utilizing more primary colours[14,15]. We, therefore, aimed for "super-multicolour" fluorescence labelling utilizing >3 types of XFPs.

Firstly, we screened for XFPs which are bright, monomeric, evenly distributed in mammalian neurons, excitable with conventional laser lines, and are spectrally dissimilar to each other. We identified a combination of 7 XFPs that met all the above criteria: mTagBFP2, mTurquoise2, and mAmetrine1.1 were excitable with 405 nm laser; mNeonGreen and YPet were excitable with 488 nm laser; mRuby3 and tdKatushka2 were excitable with 552 nm laser (Fig. 1b). We excluded XFPs that were prone to aggregation, photobleaching, and/or distributing unevenly in neurites. Emission signals for different XFPs were separated by diffraction gratings during confocal imaging.

When multiple types of XFPs are excited with a single laser, the emission spectra partially overlap with each other. To extract the fluorescence signals derived from a single type of XFP, we used linear unmixing[16,17]. For successful linear unmixing, the linearity of the fluorescence signal is critical. When detectors of the confocal microscope are not sufficiently linear (e.g., a photon-counting-based Leica HyD detector[18,19]), we have to correct the linearity to improve the performance of linear unmixing (Supplementary Fig. 1). Then, using HEK293T cells expressing a single type of XFPs, we determined how much signals leak into different channels. Based on this reference data, we calculated how much signal is derived from each XFPs (Supplementary Figs. 2–4). In this way, 2 or 3 XFPs excited with the same laser line were fully separated after linear unmixing (Fig. 1c). We wrote both MATLAB code and an ImageJ plugin for linear unmixing ("Methods" section). It should be noted that linear unmixing can be easily applied to images taken with conventional epifluorescence and confocal fluorescence microscopy (Supplementary Fig. 3). Thus, we can obtain fluorescence signals for up to 7 XFPs separately with conventional laser lines in confocal microscopy.

We previously reported the bright and stochastic multicolour labelling method, Tetbow[12]. With Tetbow, tTA, and TRE-XFP vectors are stochastically introduced into neurons. As each of the XFP genes are encoded in different plasmids or adeno-associated virus (AAVs), it is easy to increase the number of XFPs. Following the same condition as in the original study, we established 7-colour Tetbow in the mouse brain. Using in utero electroporation at E15, L2/3 neurons in the primary somatosensory cortex (S1) was labelled with 7-colour Tetbow (Fig. 1d). After 7-colour imaging and linear unmixing, we observed the combinatorial expression of 7 different XFPs. Both axons and dendrites were brightly labelled with these XFPs.

### 7-colour Tetbow provides superior discriminability of neurons based on colour hues

Our trichromatic colour vision can only recognize the combination of 3 channels, namely red, green, and blue. To quantitatively evaluate colour hues produced by ≥3 channels of the fluorescence signals, we need to introduce a numerical description of the colour hues that can also be extended to N-channel images. Fluorescence intensities in the N-channel images can be plotted in the N-dimensional space. After normalizing values across N channels (normalized to the maximum ROI value), we obtained vector-normalized intensity values (designated "colour vector", hereafter). The colour vector data will be plotted on the surface of the hyperplane in the N-dimensional space (Supplementary Fig. 5a). One easy way to assess the colour hue similarity is to calculate the Euclidean distance ($d$) between the colour vectors. The more similar the colour hues are, the shorter the $d$ value will be. We can also evaluate colour hue discriminability in this scheme. We can judge discriminable when the distance between the two-colour vectors is above a defined threshold value, $Th(d)$ (Fig. 2a).

Firstly, we examined whether neurons labelled with 7-colour Tetbow are more discriminable than the conventional 3-colour Tetbow. To compare the discrimination performance in the same condition, we utilized synthetic datasets. As the copy number of introduced XFP expression vectors follows a Poisson's distribution in Tetbow, we can simulate the expression patterns of XFPs[12] (Supplementary Fig. 5b). We simulated the expression profiles under the condition of an average of 0.1–6 gene copies/cell/colour. We compared synthetic data for 3- vs. 7-colour Tetbow. When an average of 2 copies/cell/colour was introduced, for example, only 93% of neuronal pairs were discriminable at $Th(d) = 0.2$ with 3-colour Tetbow; however, >99.9% of neuronal pairs were discriminable with 7-colour Tetbow at the same threshold (Fig. 2b and Supplementary Fig. 5c).

This does not mean that all of the 1000 neurons are fully separated because of the "birthday problem"[20]. We estimated the fraction of "uniquely-labelled neurons", which are separated by >$Th(d)$ from any other neurons (Fig. 2c). Using Monte-Carlo simulations with various numbers of cells, we calculated what percentage of cells would be uniquely labelled at different $Th(d)$ values. We found that 93.3% of neurons are uniquely labelled when 100 neurons are labelled (and 86.3% for 200 cells) with 7-colour Tetbow and $Th(d) = 0.2$ (Fig. 2d and Supplementary Fig. 5d).

Next, we evaluated colour discriminability with 7-colour Tetbow using the real samples. Layer 2/3 neurons in S1 were labelled with the 7-colour Tetbow using in utero electroporation. Four-week-old brain samples were cleared with SeeDB2G[13], and 7-channel images were obtained with confocal microscopy followed by linear unmixing. Fluorescence signals isolated from 2031 somata were analysed for colour vector similarity (Fig. 2e). We found that 99.7% of neuronal pairs were discriminable at $Th(d) = 0.2$ (Fig. 2f), which is much better than the 3-colour Tetbow (64.5%)[12]. In this condition, 79% out of 100 neurons can be uniquely labelled (Fig. 2g).

When analysing colour vectors in neurites, we must also take into account their fluctuations and noise in different neurites. In another simulation for "neurites" with noise (Supplementary Fig. 6), we found that when the standard deviation (SD) of the noise is ≤0.1 (which is realistic, as described later), discriminability with 7 XFPs is much higher than with 3 XFPs. For example, when 7 XFPs were expressed at 2 copies/cell/colour and the SD of the noise was 0.1, the neurons were 98.5% discriminable, which was much higher than 3 XFPs (only 91.2% discriminable). We also found that discriminability was highest (99.9% discriminable with 7 XFPs) when slightly lower copies (1 copy/cell/colour) was expressed.

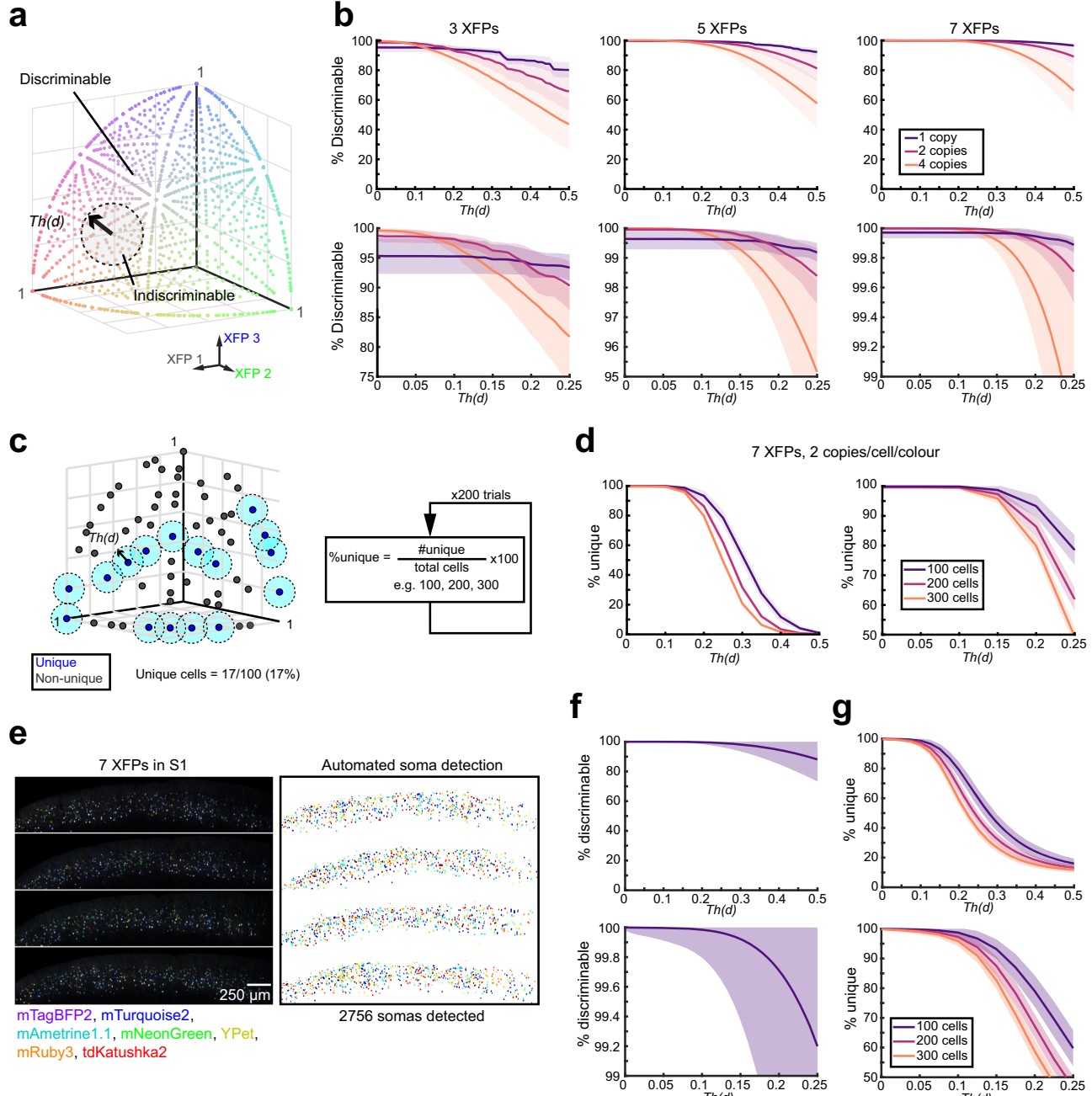

**Fig. 2 | 7-colour labelling facilitates unique labelling of more neurons than with 3-colour labelling in Tetbow method. a** Determining the number of colour vectors discriminable in 3 dimensions. Cartoon illustrating our discrimination analysis. Modelled "cells" within a threshold distance, *Th(d)*, from the chosen cell were considered indiscriminable, and those with a greater distance were considered discriminable. **b** Discrimination analysis of 10,000 modelled cells with up to 7 XFPs. Plots show the mean (bold line) ± 1 standard deviation (SD) (shaded areas) for average 1 copy (purple), 2 copies (magenta), and 4 copies (orange) per cell per colour with 3XFPs (left), 5 XFPs (middle), and 7 XFPs (right; *n* = 10,000 each). Bottom panels are zoomed in insets of the top panels. At all number of colour dimensions, 2 copies per colour per cell seemed to be optimal if the *Th(d)* was between 0.1 and 0.2. **c** Cartoon to measure the uniquely labelled cells. Cells were considered uniquely labelled if there were no neighbouring cells within a specified *Th(d)* (light blue circles). Monte-Carlo simulations (200 per condition) were

performed to calculate the percent of unique cells generated at a given number of cells, at varying concentrations, number of XFPs, and varying *Th(d)*s. **d** Modelling results at the optimum conditions (7 XFPs, at average 2 copies per colour per cell). For 100 cells, 93.3% will have a unique colour hue. Data are mean ± SD. **e** Evaluation with actual data. Four serial sections of S1 labelled with 7-colour Tetbow (*in utero* electroporation) were imaged (age, P28). Neurolucida was then used to auto-detect 2031 somata with the mean colour recorded. Z-stacked images of 2677.27 × 2369.32 × 101.9694 μm³ are shown. **f** Discrimination analysis of somata. Data are mean ± SD. **g** Fraction of uniquely labelled cells. 100, 200, or 300 somata were chosen at random (×100 simulations) and the percentage of somata that have a unique colour hue was recorded. Our data suggests that >80% out of 100 cells are uniquely labelled at *Th(d)* = 0.2. Data are mean ± SD. Source data are provided as a Source Data file.

## Strategy for automated neurite reconstructions based on colour hue similarity

As many of neurons are uniquely labelled by the combinatorial expression of 7 XFPs, we considered that we should be able to identify neurites for different neurons solely based on the colour hue information. This will be conceptually different from the existing neurite reconstruction strategy which is based on the physical continuity of the neurites (neurite "tracing")[21–27]. Thus, our strategy could overcome the limitations of the existing reconstruction methods.

Colour hue similarity can be evaluated based on the Euclidean distance between colour vectors in the 7-dimensional space as described in Fig. 2a. We initially tried to evaluate colour vector for each pixel. However, it was difficult for two reasons. Firstly, the colour vector was not stable enough at the single-pixel level. Shot noise contributed to significant fluctuations. Furthermore, the distribution of XFPs was not completely uniform at a local level (<several microns), especially in thin axons. Secondly, pixel-based analysis required a massive amount of machine power for pairwise distance calculations, hampering analysis of gigabyte-scale images. We, therefore, decided to extract colour vector data for regions of interest (ROIs) consisting of multiple pixels.

We thus needed to set ROIs for neurite fragments. There are many open-source and commercialized software to automatically detect fibrous structures such as neurites. They cannot trace neurites entirely but are good enough to detect many of the bright neurite fragments. We used one of them, Neurolucida 360[22], to automatically detect neurite fragments.

Briefly, we obtained fluorescence intensity values from each of the neurite fragments as ROIs and obtained colour vector data. We then performed unsupervised clustering of the colour vector data for all of the ROIs. If neurites from the same neuron have very similar colour hues, each cluster should represent neurites from the same neuron. In this way, we established an automated neurite reconstruction pipeline based on quantitative analysis of multiple dyes (QDyeFinder; Fig. 3).

## QDyeFinder pipeline and quality control

Here, we describe the more details of the QDyeFinder pipeline, including pre-processing, parameter setting, and quality control (QC) procedures.

Firstly, after obtaining N-channel fluorescence images, we performed linear unmixing as described in Fig. 1. To obtain N-channel signals accurately from thin neurites, it is important to minimize chromatic aberrations. Chromatic aberrations can be a problem when the objective lens is not optimized for the clearing agent. Chromatic aberration is typically more evident along z: Different colours will appear at different z positions for multicolour-labelled neurites. When necessary, we performed *post-hoc* correction of chromatic aberrations as described previously[28] (Fig. 3a).

Next, we generated channel-stacked images and automatically detected neurite fragments using Neurolucida 360. This software can detect bright neurite fragments with sufficient signal-to-noise ratios but cannot fully trace neurites with dim compartments. Existing neurite detection software, including Neurolucida 360, often makes mistakes when neurites are making branches and/or multiple neurites are crossing over along z. We, therefore, broke the detected fragments down to smaller fragments at branch and crossing-over points (Fig. 3b). These fragments were used as ROIs in subsequent analyses.

We obtained mean intensity values for N channels in each of the ROIs. As a part of QC, we examined whether all the N channels have sufficient signal-to-noise (S/N) ratios. We determined cumulative plots for signal intensities in the ROIs and background (non-ROI areas). We checked if all of the channels have sufficient S/N at the 80-100 percentile range of the intensities. We need to discard the channel when S/N in this range was <2.5, as we cannot expect reliable signals for that

channel (Supplementary Fig. 7a). There may be no or little label for that channel in that case.

We also needed to discard very short neurite fragments, as the colour vector data becomes less stable as the length decreases. We used neurite fragment data from 7-colour Tetbow samples (L2/3 neurons in S1, imaged with a 63x objective lens) to evaluate this issue. Neurite fragments were artificially broken down to smaller sub-fragments of varying lengths. To examine the relationship between length and colour vector stability, we measured the Euclidean distance ($d$) between the sub-fragments the parental (whole) fragment. We found that $d > 0.1$ when the lengths of sub-fragments become <5 µm (Supplementary Fig. 7b). We, thus, decided to consider ROIs at a length of >5 µm for this image.

We also discarded ROIs with insufficient brightness. There is a trade-off between the brightness and reliability of the colour vectors, where neurites that are dim produce unreliable colour vectors. We split each neurite fragment into sub-fragments of the minimum length and calculated the Euclidean distance ($d$) of the colour vector to that of the parental fragment. If $d > 0.2$, we defined it as an inaccurate sub-fragment. To determine a magnitude threshold, we found the smallest magnitude cut-off where less than 5% of the sub-fragments were inaccurate. We then excluded ROIs whose channel-stacked signals are below the defined threshold brightness, 0.1 (Supplementary Fig. 7c).

While we broke fragments at the branch and crossing-over points, the Neurolucida 360 software can still make mistakes for neurite identification. If the colour hue abruptly changes in the middle of the fragment, then detected fragment is most likely a "wrong" hybrid of different neurites that intersect. We, therefore, broke down the fragments further when the colour vectors in adjacent sub-fragments (segmenting by the minimum fragment length described above) of the fragment are separated by $d > 0.3$ (Supplementary Fig. 7d).

With all the ROIs with reliable signals, the mean intensity values were extracted and then vector normalized to obtain the final colour vectors. We next performed unsupervised clustering of colour vectors. Ideally, each of the cluster should represent neurite fragments from a single neuron (Fig. 3d). Clustering algorithm and its parameter setting are critical for accurate classification.

## dCrawler: A threshold distance-based unsupervised clustering

Various types of algorithms are known for unsupervised clustering (Table 1). For example, k-means clustering is a distance-based clustering and requires the number of clusters beforehand. However, it is often difficult to know the exact number of clusters (i.e., labelled neurons) in the Tetbow sample. Mean-shift clustering is a density-based clustering and requires a density kernel beforehand. However, the density of the fluorescence signals (i.e., the total length of the neurites) is not necessarily equal among neurons in the Tetbow sample.

What we want to do in the QDyeFinder is to classify the neurite fragments based on their colour hue similarity. We, therefore, developed an unsupervised clustering algorithm, named dCrawler, in which multi-dimensional vector data are classified based on a threshold distance, *Th(d)* (Fig. 4). In the dCrawler method, we aimed to classify data points into clusters, so that data points within a cluster are all within a defined *Th(d)* from their centroid position.

In the dCrawler, the first data point will be the first cluster. If a new data point is within *Th(d)* from the centroid position of the existing cluster, we included it in that cluster and updated the centroid position. If the data point was separated by >*Th(d)* from the centroids of any existing clusters, we assigned a new cluster to the data point. We repeated this process until all of the data points were assigned to clusters (Fig. 4a–f). To avoid a primacy bias, we then re-allocated points to their closest centroid and updated the centroid positions (Fig. 4g–i). Lastly, we merged clusters whose centroids are located within a defined distance (Fig. 4j–l). All the

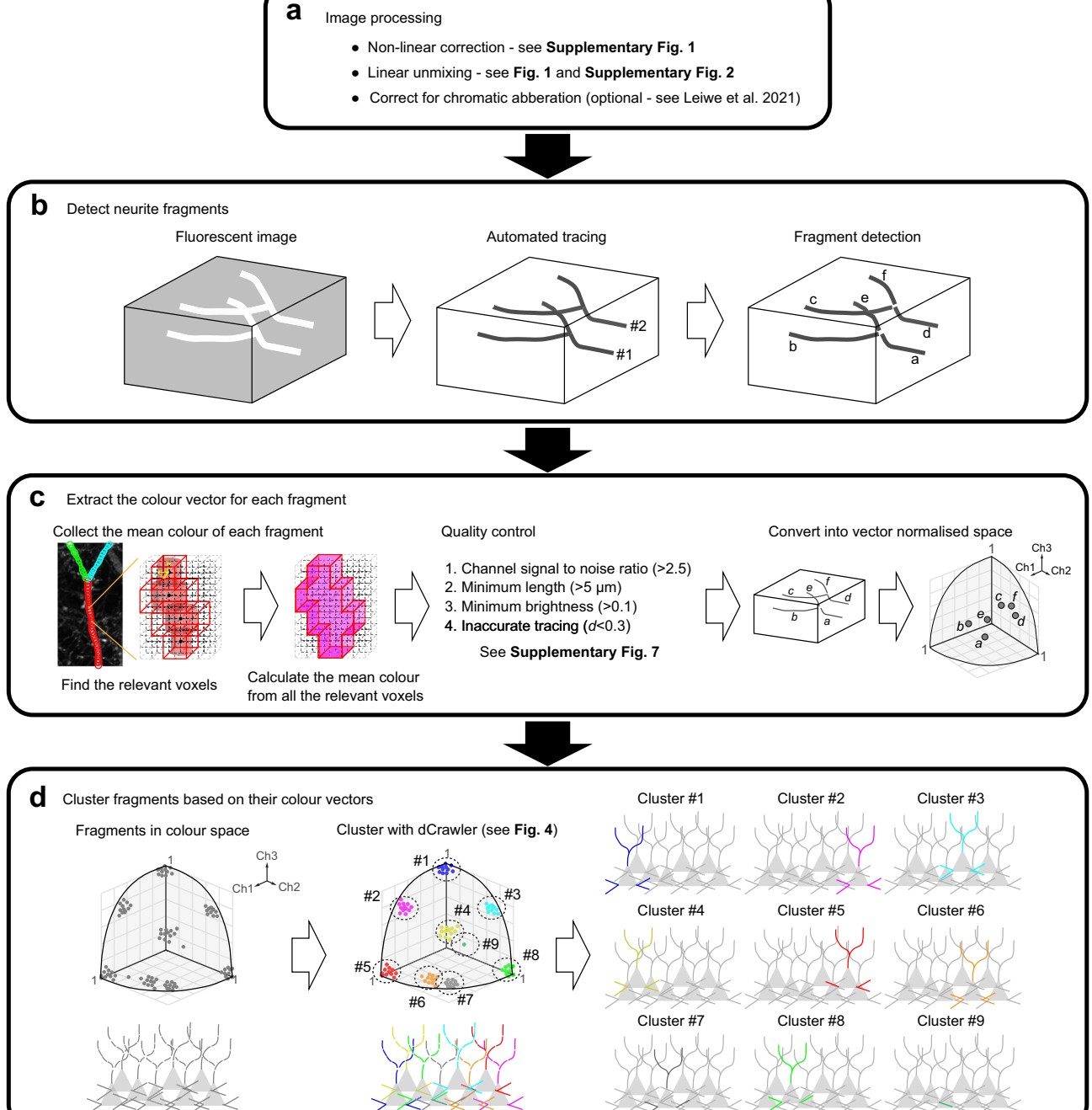

**Fig. 3 | QDyeFinder pipeline for automated reconstruction of neurites. a** Image processing. Firstly, the images were non-linearly corrected (Supplementary Fig. 1), then the images were linearly unmixed (Fig. 1c and Supplementary Fig. 2). If axial chromatic aberration correction was necessary, it was performed as described in Leiwe et al.[28]. **b** Neurite detection with Neurolucida 360 (MBF Biosciences). Fluorescence images were loaded into Neurolucida 360, to automatically detect neurites using the directional kernels method. However, mistracing frequently occurred in densely labelled images, especially at branch and crossing points. Therefore, detected neurites are split into fragments at branch and crossing points (right). Note that somata were excluded in this step, as they are too bright, and their signals are beyond the linear range. Our QDyeFinder pipeline supports data in.swc format, which is commonly used for neurite reconstruction. To use Imaris Filament Tracer (Oxford Instrument) for automated neurite fragments, see Supplementary Note 1. **c** Once the fragments have been detected, the relevant voxels are identified, and the mean pixel intensity is calculated for each channel. After quality control (Supplementary Fig. 7), fragments are represented in vector normalised space (colour vectors). **d** Colour vectors for the fragments are then clustered based on *Th(d)* with our clustering algorithm dCrawler (left and middle). Individual clusters can then be plotted to identify neurons from individual fragments (right).

processes were repeated until all the data points are assigned to clusters. To avoid infinite looping, the merging threshold begins to be gradually reduced (*Th(d)*\*0.99) for each loop after 20 loops. Eventually, we can obtain clusters whose data points are within *Th(d)* from their centroids (Fig. 4l).

Once we have defined an appropriate *Th(d)* value based on experimental data (e.g., small-scale data under similar conditions), we should be able to classify numerous neurite fragments based on the similarity of their colour vectors. Under the appropriate *Th(d)* value, each of the clusters should represent neurites from a single neuron.

**Table 1 | Comparison of unsupervised clustering algorithms**

| Clustering Algorithm | k-means clustering | Mean shift clustering | DBSCAN | dCrawler |
|---|---|---|---|---|
| What it measures | Distance | Density | Density and distance | Distance |
| Input required | Number of clusters | A density kernel | Minimum points and distance | A threshold distance |
| Advantage | Standard and fast | Outliers have very little effect | Considers both the density and distance of the points | Input does not require any assumptions on the spread of data |
| Disadvantage | We do not know the number of final clusters | The density of clusters may be variable | The clusters may be unevenly spread over space | Can produce too many clusters |

K-means clustering is commonly used for unsupervised clustering (also in a previous study[30]). However, it is often difficult to know the exact number of clusters (neurons) in real samples. Similarly, mean shift clustering may not be very useful because it is difficult to set the density kernel based on the microscopy images; also, different neurons may not produce a similar number of ROIs within the imaged area. DBSCAN is not useful for QDyeFinder, because multiple clusters representing different neurons can be easily fused. dCrawler is much simpler because we only need to set the $Th(d)$, which is easy to determine based on example images; it was typically ~0.2 in our examples using 7-colour Tetbow. dCrawler can potentially produce too many clusters with few data points (derived from noise signals); however, clusters with few data points can be discarded as useless.

## Evaluating the performance of QDyeFinder

To evaluate the performance of QDyeFinder, we compared the performance with conventional manual tracing. We used a 100-μm-thick 4-week-old mouse brain sample, in which layer 2/3 neurons were labelled with 7-colour Tetbow. After obtaining 7-channel images with a confocal microscope with a 63x objective (numerical aperture (NA) = 1.30), we found 35 neuronal somata expressing at least one of XFPs. To obtain the "ground-truth" data, we manually traced all the neurites (mostly dendrites) from the 35 neuronal somata (Fig. 5a). The traced neurites were then broken into smaller fragments according to the standard procedure in QDyeFinder (Fig. 3b). We examined whether dCrawler with an appropriate $Th(d)$ value can accurately classify the neurite fragments for different neurons.

Using manually traced ground truth data, we checked whether the colour hues are consistent across all the neurite fragments. We examined the distribution of colour vectors and found that for most neurons, the distance ($d$) to the mean value of all the neurite fragments belonging to that neuron (centroid) is less than 0.2 for most of the neurons (Fig. 5b, Supplementary Fig. 8a, b, and Supplementary Data 1). UMAP plotting of all the 679 neurite fragments demonstrated clear and discrete clusters, each of which represents neurite fragments from the same neuron (Fig. 5c). We next optimized the $Th(d)$ value for the dCrawler. We used the F1 score to evaluate correct classification based on the ground truth data. We performed dCrawler with different $Th(d)$ values and calculated the median F1 score across neurons. For this particular sample, the $Th(d)$ that demonstrated the highest median F1 score (0.971) was a $Th(d)$ of 0.2. The dCrawler at $Th(d) = 0.2$ identified 42 clusters (Fig. 5e). A side-by-side comparison of the manually traced neurons (blue) and identified clusters (green) demonstrated a high degree of consensus (Fig. 5f, Supplementary Fig. 5c, and Supplementary Data 2). Thus, the performance of neurite reconstruction with QDyeFinder was overall comparable to manual tracing for this sample. The remaining errors (pseudo-positives and negatives) are likely due to poor colour hue representation, errors in manual tracing, and/or the birthday problem (Fig. 2f).

## Automated reconstruction of dendrites and axons with QDyeFinder

We next tried the QDyeFinder pipeline for an independent brain sample without a ground truth data. Again, layer 2/3 neurons were labelled with 7-colour Tetbow using *in utero* electroporation. We took a fluorescence image spanning a $581.53 \times 454.32 \times 290.41$ μm³ volume using a ×20 objective (NA = 0.75; Fig. 6a). A total of 290 labelled neuronal somata were found within this volume. In addition, it should contain neurites from many more neurons including those extending from outside this volume. The automatic detection program identified 15,174 neurite fragments (median fragment length, 12.1 μm; interquartile range, 7.9-20.3 μm) (Fig. 6b), many of which should be dendrites. The dCrawler at $Th(d) = 0.2$ identified 302 clusters (Fig. 6b). In

many of the clusters, the identified neurite fragments were spatially clustered, suggesting that each cluster corresponds to neurites from one or a few neurons (Fig. 6c–e). We also ran the dCrawler at different $Th(d)$s where the F1 score was above 0.9, ($Th(d) = 0.13$, 0.15, 0.175, 0.225, 0.25, and 0.26). To evaluate the effectiveness of the clustering, each cluster was allocated into one of three groups: where a single neuron is visible (Fig. 6d, left panels, >90% of fragments seem to belong to a single neuron), where few fragments are visible, but no neuron is visible (Fig. 6d, middle panels, <30% of fragments for a neuron seemed to be detected), and finally a cluster where the fragments belong to two or more neurons (Fig. 6d, right panels). The percentage of clearly visible single neuron clusters increases up to the optimized $Th(d)$ of 0.2, and then begins to decrease. We sometimes observed occasions where multiple clusters actually represented the same neuron, especially at lower $Th(d)$ values (Supplementary Fig. 9).

We next examined whether QDyeFinder can accurately reconstruct axons. Reconstructing long-range projecting axons is more challenging than dendrites, as axons are much longer and thinner. We tested mitral/tufted cells in the olfactory bulb, which project millimetre-long axons to the olfactory cortices. A typical mitral/tufted cell projects a single axon within the lateral olfactory tract (LOT) and extends several collaterals towards various cortical regions. AAV-CAG-tTA and AAV-TRE-XFP were locally injected into the olfactory bulb to brightly label a limited number of neurons (<100). Three weeks after the injection, the LOT and olfactory cortices were imaged with confocal microscopy (20x objective, NA = 0.75; Fig. 7a). In a pilot trial with a small-scale sample, we compared the results of QDyeFinder and manually traced ground truth and obtained F1 score of 0.955 at $Th(d) = 0.2$ (Supplementary Fig. 10). Then, we tested large-scale images ($2629.5 \times 1636.4 \times 437.3$ μm³). A total of 4230 axonal fragments were automatically detected (median fragment length, 24.1 μm; interquartile range, 14.4–50.5 μm; Supplementary Fig. 11). We classified the type of the clusters (single neuron, few fragments, and multiple neurons) at varying $Th(d)$s. We found that a $Th(d)$ of 0.25 produced the highest number of single neuron clusters (Supplementary Fig. 11c). Therefore, we evaluated the fragments at $Th(d) = 0.25$. A dCrawler at $Th(d) = 0.25$ identified 88 clusters, with 14 clusters that clearly label axons for a single neuron (Fig. 7b, c and Supplementary Fig. 11b). Many of the clusters included axons spanning a millimetre scale. Thus, super-multicolour labelling combined with QDyeFinder can be used for the analysis of long-range axonal projection.

We also analysed fine axonal collaterals and presynaptic boutons using a high NA objective lens. Using the same procedure as in Fig. 7, we labelled axons of mitral/tufted cells and imaged their collaterals in the piriform cortex using a higher magnification objective lens (×63 objective, NA = 1.30). Axons from single neurons were successfully reconstructed, although the fluorescence signals of fine axons are often fragmented (Fig. 8), which is difficult to reconstruct with existing auto-tracing software. Thus, we can reconstruct fine neuronal

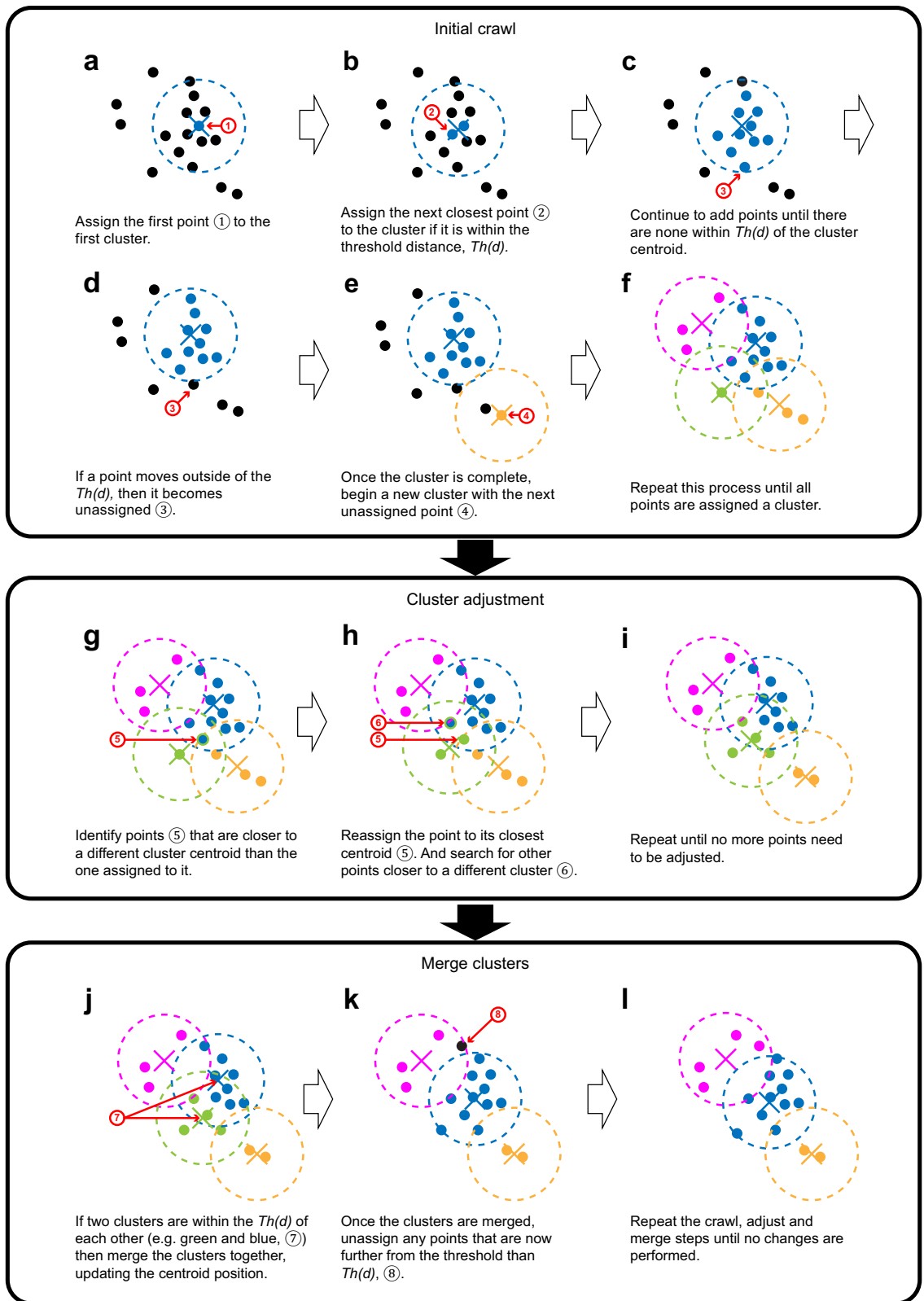

**Fig. 4 | Schematic for an unsupervised clustering method, dCrawler.** dCrawler has three main steps. The first step is an initial crawl (**a–f**) where a centroid identifies the nearest non labelled point within a specified distance, *Th(d)*, and then updates its position. This continues until all points have an allocated cluster (**f**). Then, the adjustment phase occurs (**g–i**) where each point is re-allocated to its nearest centroid, and then the centroid positions are updated. This is then repeated until all the points are associated with their closest centroid (**i**). Next, the merge phase begins (**j–l**), where if any centroids are within *Th(d)* of each other, they are then merged, with the any points that are outside of the cluster being unassigned (e.g. **k**). Finally, the crawl, adjust, and merge steps are repeated until a stable solution is reached. See Supplementary Note 2 and Supplementary Movie 1 for more details.

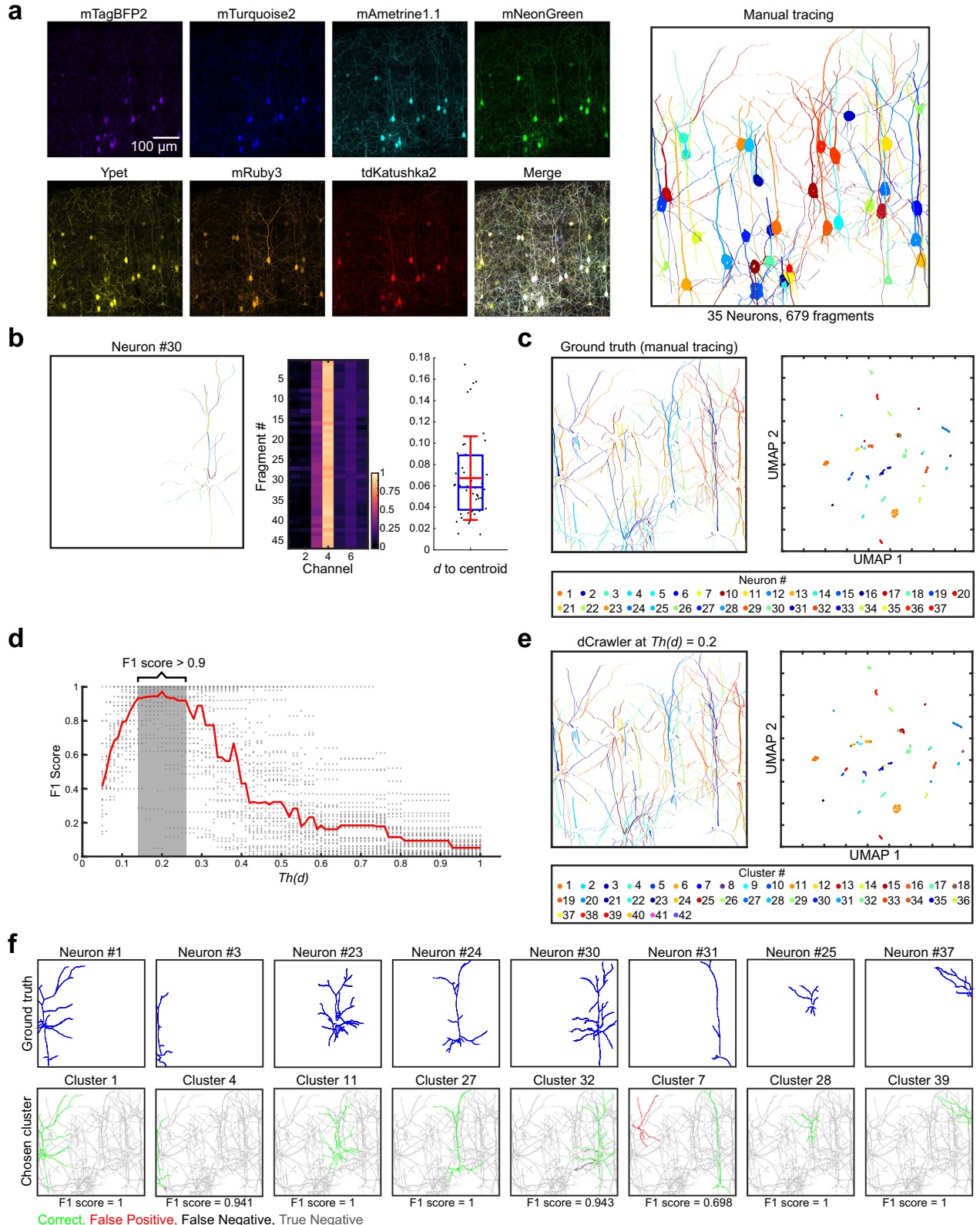

**f** Ground truth / Chosen cluster panels:

Neuron #1 — Cluster 1 — F1 score = 1
Neuron #3 — Cluster 4 — F1 score = 0.941
Neuron #23 — Cluster 11 — F1 score = 1
Neuron #24 — Cluster 27 — F1 score = 1
Neuron #30 — Cluster 32 — F1 score = 0.943
Neuron #31 — Cluster 7 — F1 score = 0.698
Neuron #25 — Cluster 28 — F1 score = 1
Neuron #37 — Cluster 39 — F1 score = 1

Correct, False Positive, False Negative, True Negative

processes with the QDyeFinder pipeline if we image them with sufficient resolution.

**Reconstruction across multiple brain slices**

High-resolution fluorescence imaging is essential for the reconstruction of fine neurites. However, the reconstructions of large volume are challenging, because high magnification objective lenses typically have short working distance (typically ≤300 μm for NA ≥ 1.3). This means that brain samples have to be cut into multiple thin slices. However, the reconstruction of neurites across multiple brain slices is extremely challenging for existing neurite reconstruction methods, where reconstruction is performed based on the physical continuity of neurite signals. Our QDyeFinde has a unique advantage over previous methods in that the reconstruction is performed

**Fig. 5 | Comparison of QDyeFinder versus ground truth with manual tracing.**
**a** L2/3 neurons in S1 were labelled with 7-colour Tetbow via *in utero* electroporation
(age, P28) (left). 35 neurons were then manually traced via Neurolucida 360. They
were then split into fragments (no branches). Each fragment is represented by a
different colour (right). Somata are also highlighted, but were excluded from the
analyses. Image taken with a ×63 objective lens (NA = 1.30). Z-stacked images of
$511.36 \times 512.99 \times 47.99$ μm³ are shown. **b** A representative example neuron showing
the relevant fragments (left), the vector normalised colour vector of each trace
(middle), and the Euclidean distance to the mean for the neuron (right). Red bars
indicates mean ± SD and blue box plots indicate median ± interquartile range (IQR)
of *d* to the centroid of the colour vectors for all fragments ($n = 47$ fragments). See
Supplementary Data 1 for all the results. **c** The ground truth of each neuron and
their neurite fragments in physical space (left), and in UMAP-reduced colour space

(right). **d** The optimum threshold was calculated by running the dCrawler at a *Th(d)*
ranging from 0.05 to 1, and calculating F1 score for each neuron at each *Th(d)* (gray
dots). The median F1 score for each *Th(d)* is also displayed (red line). The optimum
*Th(d)* was calculated to be 0.2. **e** The dCrawler clustering of the neurite fragments at
the the optimum *Th(d)*, shown in physical space (left), and in UMAP-reduced colour
space (right panel). **f** Representative neurons (ground truth in blue lines, top row)
paired to their best cluster. Fragments in both the ground truth and the dCrawler
cluster (bottom row) are considered correct (green), those only in the dCrawler
cluster are a false positive (red), those in the ground truth only as a false negative
(black), and those in neither the ground truth or dCrawler cluster as a true negative
(grey). See Supplementary Data 2 for all the results. Source data are provided as a
Source Data file.

based on the colour hue similarity of neurites, rather than the phy-
sical continuity.

We, therefore, tested whether we can identify neurites that are
spanning multiple physical brain sections. We labelled olfactory bulb
neurons with 7-colour Tetbow and imaged 4 consecutive brain slices
(100 μm thick). After processing all the four images together using
QDyeFinder pipeline, we identified multiple apparent single neurons
spanning across multiple slices (Fig. 9). Thus, QDyeFinder allows for
reconstruction of neurites across multiple brain slices much easier
than before.

## Discussion

During the past years, large-scale image acquisition is becoming easier
for EM and LM-based connectomics. However, circuit reconstruction is
still a laborious and technically challenging. Existing neurite recon-
struction pipelines are based on "tracing" of physically continuous
structures. Thus, slight damage to the images would have a deleterious
impact for successful circuit tracing. The error rates will exponentially
increase as the distance of the traced neurites increases, hampering
large-scale reconstructions.

In this study, we developed a conceptually different type of
neurite reconstruction pipeline based on super-multicolour labelling
and QDyeFinder. In this strategy, we identified neurite fragments only
based on their colour hue information. In other words, each of the
neurite fragments have a unique fluorescent "barcode"[4]. We, therefore,
do not need to care about the physical continuity of the neurite frag-
ments. This means that the physical distance does not limit the accu-
racy of the reconstruction, as in previous methods (Supplementary
Table 1). Accordingly, QDyeFinder outperforms the state-of-the-art
auto-tracing software for densely labelled samples (Supplementary
Fig. 12). We do not even need to take continuous images of neurites as
long as the neurons are labelled with unique and consistent colour
hues (Fig. 9). Moreover, QDyeFinder is fully automated, and excludes
any human biases in reconstruction. This is conceptually similar to a
DNA barcode-based connectomics, such as MAPseq and BARseq[20,29].
However, our fluorescent protein-based approach is more useful for
morphological analyses in 3D. Morphological information should tell
us a lot about how neurons receive synaptic inputs, integrate them,
and send the output to other neurons. Combined with a cell type-
specific Cre driver, we can analyse morphological variation within the
same cell type. We can even analyse synaptic structures, such as den-
dritic spines and axonal boutons, although this is time-consuming
when imaged at the highest resolution (e.g. NA ≥ 1.3) using SeeDB2
(Fig. 8)[12,13].

With 7 XFPs, we can differentiate 99.9% of neuronal pairs. How-
ever, the practical limit of neuronal numbers is -100 due to the
birthday problem (Fig. 2g). For example, clusters frequently contained
multiple neurons in Fig. 6, where >290 neurons were labelled. In
contrast, a majority of the clusters corresponded to single neurons
when <100 neurons were labelled (Figs. 5 and 8 and Supplementary
Fig. 10). Thus, it is critical to limit the number of labelled neurons to

identify majority of neurons with unique colour hues. Another critical
factor for successful application is the consistency of the colour hues.
In this study, we carefully screened for XFPs that were bright and
evenly distributed in neurites. The ROI (neurite fragments) should
cover a sufficient length (>5 μm) of neurites for stable colour repre-
sentation (Fig. 3c and Supplementary Fig. 7b). The linearity of the
detectors is also critical (Supplementary Fig. 1). Notably, the colour
hues at soma, dendrites, and axons were different due to localization
bias of XFPs and different signal/background ratios under different
brightness. Dendrites and axons should also be analysed separately
because of the large difference in brightness between them. To suc-
cessfully analyse thin axons, it is also important to label neurons
brightly and use high NA objectives. To obtain consistent colour hues,
it is important to avoid photo-bleaching. Tissue clearing is also
important as blue signals scatter more in thick brain samples. We used
SeeDB2G, because fluorescent proteins are bright and stable in this
clearing agent[13].

With our QDyeFinder, unsupervised clustering with dCrawler was
critical. dCrawler has unique advantages over existing clustering
algorithms (Table 1) and should be useful for various kinds of high-
dimensional data analysis, not only in biology (e.g., cell typing with
highly multiplexed in situ hybridization and/or antibody staining), but
also in other fields. In a previous study aimed at automated analysis of
Brainbow data, colour vectors in superpixels were classified using
k-means clustering[30]. However, k-means clustering requires users to
estimate the number of clusters, which may be more challenging than
estimating the optimal *Th(d)*. Furthermore, we found that dCrawler
performs better than k-means clustering under the best parameter
setting when implemented in our QDyeFinder pipeline (Supplemen-
tary Fig. 13). We, therefore, conclude that dCrawler is more useful for
classifying colour vectors in Tetbow data.

Mean-shift clustering is not appropriate for our purpose because
the density of data points (i.e., the number of fragments) will be vari-
able for each neuron. DBSCAN is a greedy algorithm; clusters can easily
expand over the range of thresholds if any points lie between clusters
(Supplementary Fig. 14a). DBSCAN is powerful when the data is not
normally distributed and the clusters are distinct (Supplementary
Fig. 14b). However, QDyeFinder should be better suited for sorting
spectral data such as colours in images (Supplementary Fig. 14c).

The appropriate *Th(d)* value should be determined based on
the stability of colour hues within a neuron and number of labelled
neurons in the sample. Based on our optimization for dendrites (Figs. 5
and 6, Supplementary Fig. 8), axons (Figs. 7 and 8 and Supplementary
Figs. 10 and 11), and sub-fragment analyses in axons (Supplementary
Fig. 7c), we expect that the best *Th(d)* value is in the range of 0.20-0.25
when multicolour images are carefully acquired. In this case, we can
isolate individual neurons reasonably well when <100 neurons are
labelled. However, the optimal *Th(d)* value may increase when labelling
and image quality (e.g., consistent labelling, signal-to-noise ratio, and
linearity) are poor. Like existing tracing methods, image quality is
critical to successful neurite reconstruction.

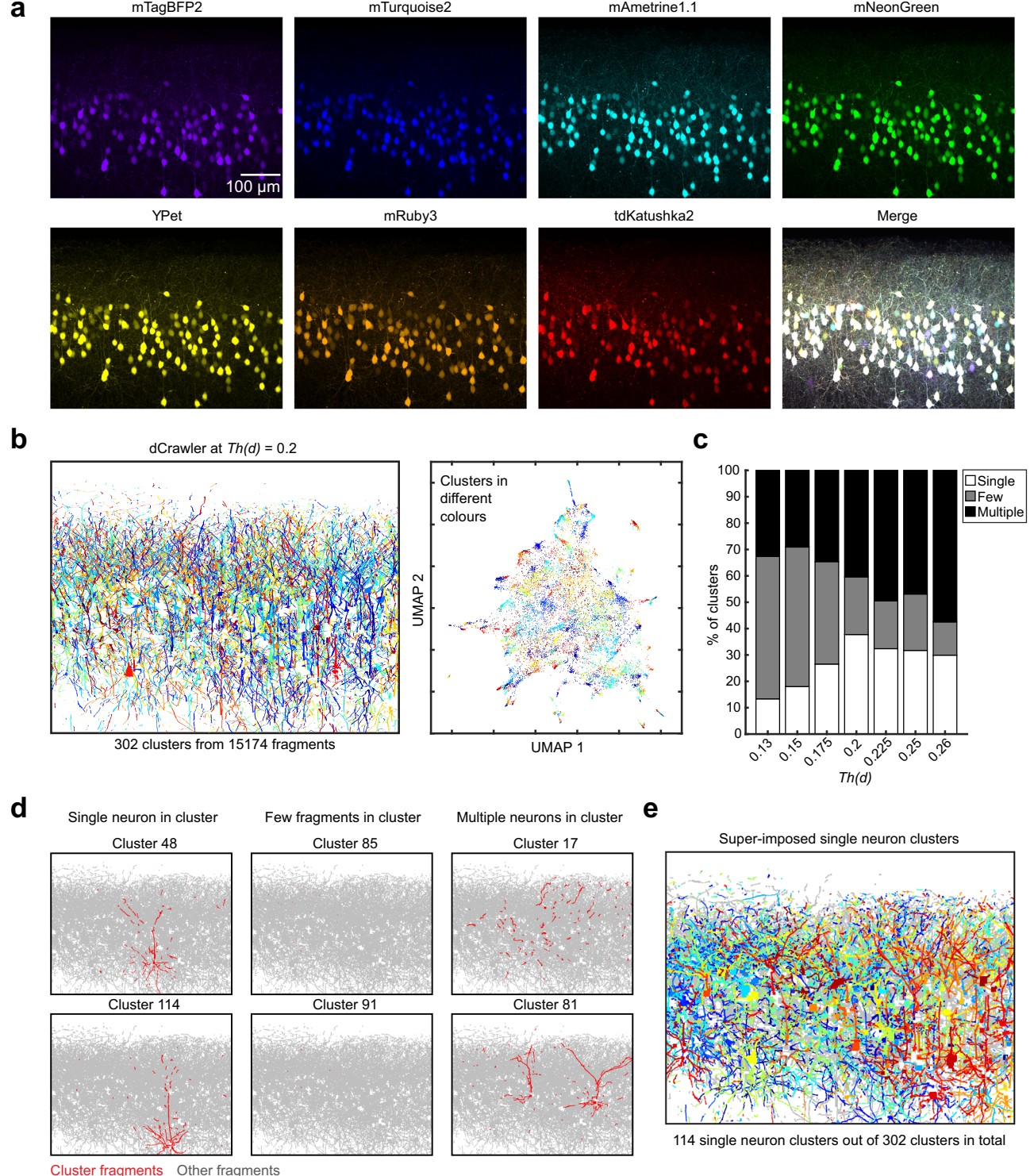

**Fig. 6 | Fully automated reconstruction of densely labelled neurons with 7-colour Tetbow in the cerebral cortex. a** L2/3 neurons in S1 labelled with 7-colour Tetbow (*in utero* electroporation). Z-stacked images of 581.53 × 454.32 × 290.41 μm³ are shown. Note that the data are from a representative result from four independent experiments with similar results. **b** 15,174 fragments were then automatically detected using Neurolucida 360 and put through our processing pipeline which detected 302 clusters after clustering at *Th(d)* = 0.2 (left panel). Each fragment is represented by their unique cluster colour in both the fragment plot (left panel) and the UMAP plot (right panel). Colours correspond to each cluster. **c** Classification of dCrawler clusters at a range of *Th(d)*. Clusters were grouped by whether a cluster contained a single neuron, multiple neurons, or few fragments. As with the manually traced data the best results were with the optimum *Th(d)* of 0.2. The optimum *Th(d)* of 0.2 provided the best percentage of single neuron clusters (37.75%). **d** Two representative example clusters are provided for each of the classification groups, fragments belonging to the cluster (red), and the remaining clusters (grey) are shown. **e** Single neuron clusters (114 clusters) are shown. Source data are provided as a Source Data file.

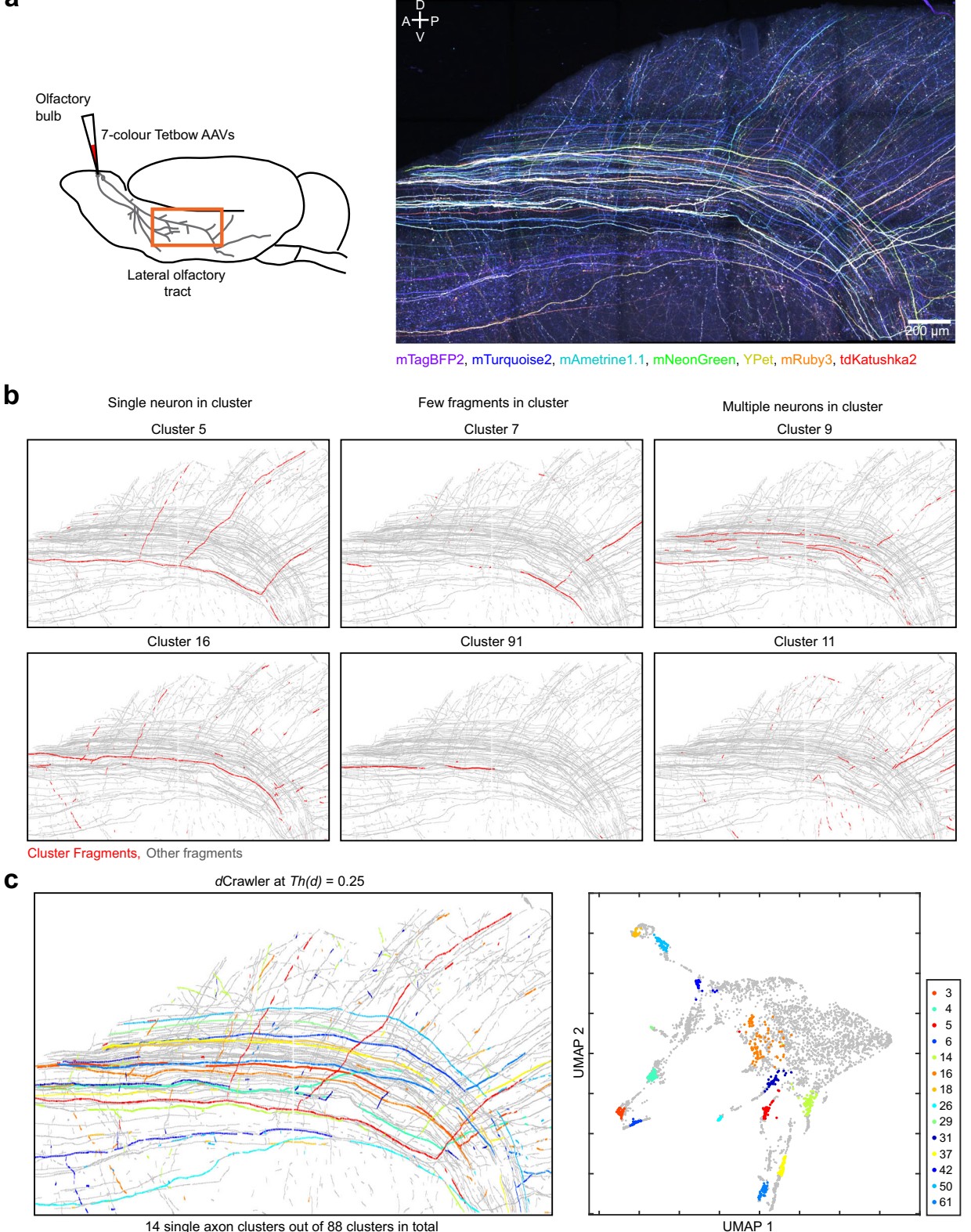

**Fig. 7 | Fully automated reconstruction of mitral/tufted cell axons at a millimetre scale. a** A 3 × 5 tiled image (A z-stacked image of 2629.55 × 1636.36 × 437.31 μm³) of the lateral olfactory tract and olfactory cortex, with mitral and tufted cell axons labelled with 7-colour Tetbow (13-week-old male). Mitral/tufted cell-specific Pcdh21-Cre mice and FLEX-tTA AAV was used to label mitral/tufted cells with AAV-TRE-XFP. Images taken with a ×20 objective. 7 XFP images are merged after linear unmixing. Note that the data are from a representative result from four independent experiments with similar results. **b** Representative clusters of the three types after QDyeFinder at *Th(d)* = 0.25. Clusters were either clearly single neuron, containing few fragments, or containing axons from multiple neurons. See supplemental data for more details. **c** All 14 single neuron clusters (left panel) and their positions in UMAP space (right panel). Colours correspond to each cluster. See Supplementary Fig. 11 for individual plots.

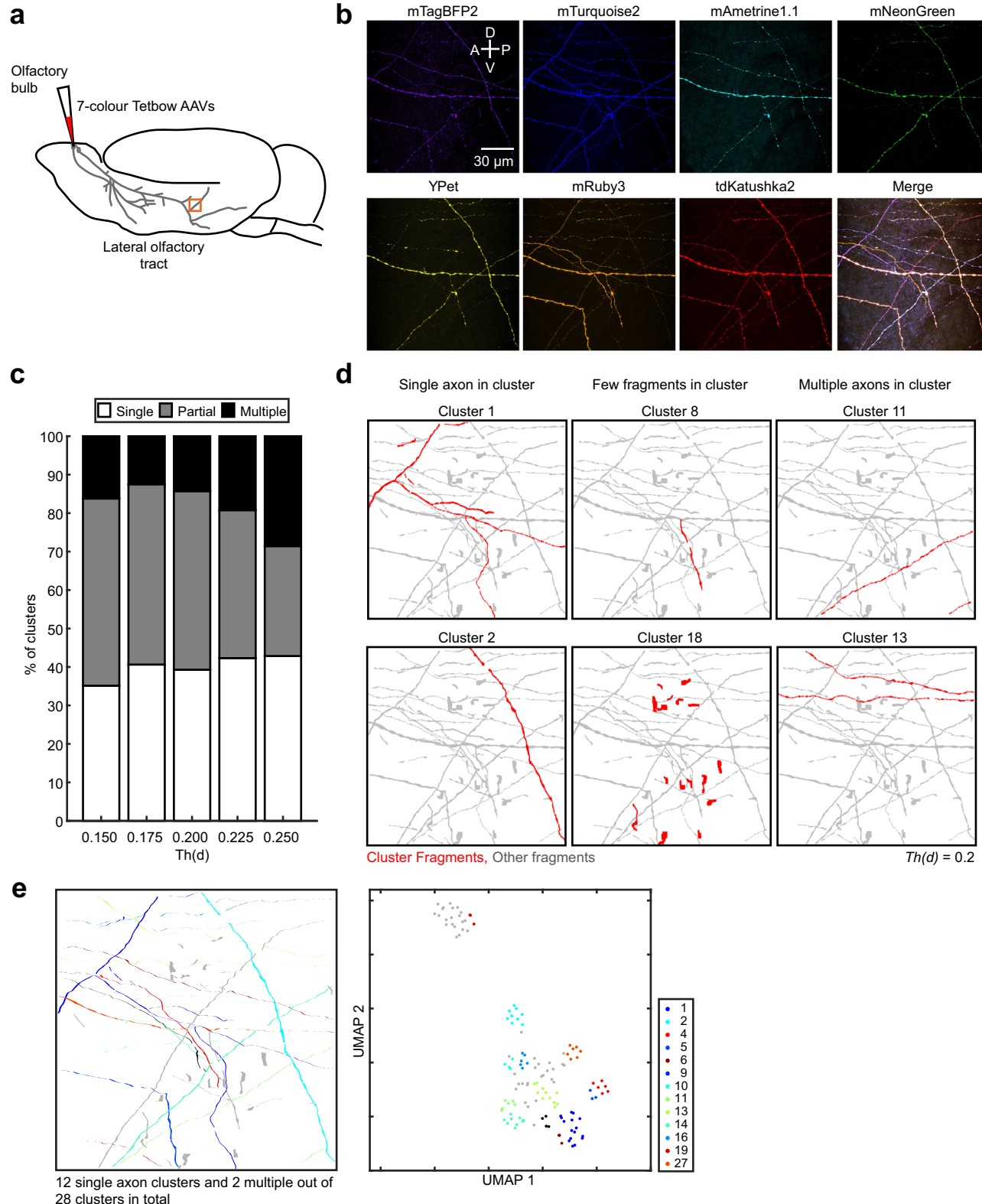

Cluster Fragments, Other fragments

*Th(d)* = 0.2

12 single axon clusters and 2 multiple out of 28 clusters in total

In the future, we may be able to differentiate more neurons with unique colour hues, if we can utilize more XFPs. Chemical tags are also potentially useful to expand spectral variations[12]. While 7 XFPs are close to the upper limit due to spectral overlap, we may be able to utilize fluorescent lifetime to further differentiate more XFPs[31,32]. If we can use peptide-based barcodes and detect them serially by antibodies or de novo designed peptide binders[33–35], we should be able to massively increase the number of unique labels[36].

In this study, we utilized super-multicolour imaging and QDyeFinder for neurite identification. In the future, we should be able to use a similar approach for time-lapse imaging. Like neurite reconstruction in 3D, cell tracking is challenging in a single-colour time lapse images. Super-multicolour labelling with QDyeFinder should facilitate more accurate cell tracking. While we ignored spatial information in our QDyeFinder pipeline, we can, of course, consider both spatial and colour hue information for more accurate neurite/cell tracking. Our

**Fig. 8 | Reconstruction of fine axonal collaterals. a** Cartoon schematic, we injected 7-colour Tetbow AAVs into the olfactory bulb with the same methodology as Fig. 7. This time, we imaged the branches that innervate the posterior piriform cortex with a x63 objective to obtain a higher resolution. The mouse was 16-week-old male. **b** Individually adjusted maximum intensity projections of the x63 image for each of the XFPs used. **c** Summary of the percentages of clusters that were determined to include a single axon, a few fragments, or multiple complete axons. A *Th(d)* between 0.175 and 0.225 appears to be best. **d** Representative clusters of our QDyeTracer pipeline produced 28 clusters at a *Th(d)* of 0.2. Example clusters

that display a single axonal tract (left panels), a few fragments (middle panels), and multiple axons (right panels). Cluster 18 does not represent axons (maybe auto-fluorescence). Note that multiple axons found for a cluster may in fact originate from the same neuron because we imaged a limited area in this case. **e** Illustration of all the single and multiple axon clusters (left) and in UMAP reduced colour space (right) Poorly defined clusters are labelled in grey. Note that most of the poor clusters contained very few fragments (categorized as "partial"; most likely auto-fluorescence or noise signals). Source data are provided as a Source Data file.

quantitative approach should be useful for highly multiplexed fluorescence imaging for various kinds of applications.

## Methods

### Plasmids
XFPs were initially selected based on the information in FPbase (https://www.fpbase.org/)[37]. Candidate XFPs were further evaluated based on the brightness and distribution in neurons. pCAG-tTA (Addgene #104102), pAAV-SYN1-tTA (Addgene #104109), pBS-TRE-mTurquoise2-WPRE (Addgene #104103), pAAV-TRE-mTurquoise2-WPRE (Addgene #104110) were described previously[12]. mTagBFP2[38] was generated from TagBFP (Evrogen). mNeonGreen[39] was obtained from Allele Biotechnology. YPet[40] was amplified by PCR from pCAGGS-RaichuEV-Rac[41]. Plasmids encoding, mAmetrine1.1[40] (Addgene #18084), mRuby3[42] (Addgene #74234), and tdKatushka2[43] (Addgene #30181) were obtained from Addgene. Each XFP gene was PCR-amplified and subcloned into pBS-TRE or pAAV-TRE vector[12]. pBS-TRE-mTagBFP2-WPRE (Addgene #193332), pBS-TRE- mAmetrine1.1-WPRE (Addgene #193333), pBS-TRE- mNeonGreen-WPRE (Addgene #193334), pBS-TRE-YPet-WPRE (Addgene #193335), pBS-TRE-mRuby3-WPRE (Addgene #193336), pBS-TRE-tdKatushka2-WPRE (Addgene #193337), pAAV-CAG-tTA (Addgene #193338), pAAV-FLEX-tTA (Addgene #149363), pAAV-TRE-mTagBFP2-WPRE (Addgene #193339), pAAV-TRE-mAmetrine1.1-WPRE (Addgene #193340), pAAV-TRE-mNeonGreen-WPRE (Addgene #193341), pAAV-TRE-YPet-WPRE (Addgene #193342), pAAV-TRE-mRuby3-WPRE (Addgene #193343), and pAAV-TRE-tdKatushka2-WPRE (Addgene #193344) were generated in this study and are available at Addgene.

### Mice
All animal experiments were reviewed and approved by the Institutional Animal Care and Use Committee of Kyushu University. ICR mice (both male and female, Japan SLC, RRID: MGI: 5652524) were used for *in utero* electroporation (Figs. 1d, 2e, 5a, and 6a) (analysed at P21-28). C57BL/6 N mice (Japan SLC, RRID: MGI: 5658686) and mitral/tufted cell-specific Pcdh21-Cre[44] in C57BL/6 N background were used for AAV experiments (7-13 week-old male and female). Mice were kept under a consistent 12 h light/12 h dark cycle (lights on at 8 am, and off at 8 pm), ambient temperature 22-26 °C, and humidity 40-70%. Mice had free access to food (CLEA Rodent Diet CE2).

### In vitro experiments (HEK cells and spectrum data)
HEK293T cells were cultured in high-glucose DMEM (044-29765, FUJIFILM-Wako) with 10% FBS, and 1% penicillin/streptomycin (FUJIFILM -Wako) under the humidified conditions in 95% air and 5% CO2 at 37 °C. For spectral measurements, pCAG-tTA2 (0.35 ug) and one of pBS-TRE-XFP-WPRE (0.9 ug) vectors were transfected to 50-80% confluent HEK293T cells in 35-mm dish using PEI-MAX (25 mM, 5 μL/dish; #24765, Polysciences, Inc.). Twenty-four hours after transfection, cells were washed twice with 1 mL of PBS, and collected with 1 mL of PBS. Cell suspensions were transferred into a glass cuvette. Excitation and emission spectra were quantified using a fluorescence spectrophotometer (F2700, Hitachi). For each XFP, spectra were measured 5 times in 0.5 nm wavelength increments and averaged. Excitation (ex) and

emission (em) wavelengths to determine the spectral curves (ex, em) were as follows (in nm): mTagBFP2 (400, 460), mTurquoise2 (430, 505), mAmetrine1.1 (400, 535), mNeonGreen (465, 540), YPet (490, 550), mRuby3 (530, 610), tdKatushka2 (550, 650). To obtain reference images for unmixing, single colour labelled cells were cultured on a poly-D-lysine coated 35-mm glass-bottom dish (#P35G-0-14-C, MatTek). Twenty-four hours after transfection, cells were fixed with 4% paraformaldehyde, washed twice with PBS, and treated with SeeDB2G solution[13].

### *In utero* electroporation
*In utero* electroporation was performed as described previously[12]. To label cortical layer 2/3 neurons in S1, 1-2 μL of plasmid solutions (0.1 μg/ μL of pCAG-tTA and 0.1 μg/ μL of pBS-TRE-XFP-WPRE each) were injected into the lateral ventricle at E15 and electric pulses (a single 10-ms poration pulse at 72 V, followed by five 50-ms driving pulses at 42 V with 950-ms intervals) were delivered toward the medio-lateral axis of the brain with forceps-type electrodes (LF650P5, BEX) and an electroporator (CUY21EX, BEX).

### AAV production
AAV vectors (serotype DJ) were generated using the pHelper (AAVpro Helper-free system, #6673, Takara), pAAV-DJ (Cell Biolabs), and the AAVpro 293 T cell line (#632273, Takara) following the manufacturers' instructions. AAV vectors were purified using the AAVpro Purification Kit All Serotypes (#6666, Takara). Viral titers were measured using AAVpro Titration Kit (#6233, Takara) or THUNDERBIRD SYBR qPCR Mix (QPS-201, TOYOBO) with StepOnePlus system (ThermoFisher) or QuantStudio3 real-time PCR system (Applied Biosystems).

### AAV injections into the olfactory bulb
To infect the Tetbow AAV vectors, C57BL/6 N mice (Fig. 9) and Pcdh21-Cre mice[44] (Fig. 7 and Supplementary Figs. 10 and 11) at age 8–13 W were used. Mice were anaesthetised with i.p. injection of ketamine (100 mg/kg) and xylazine (10 mg/kg). The hair between the mouse's head and the ears was removed, then the mouse was fixed onto a stereotaxic frame (Stoelting). The scalp and underlying connective tissue was then removed. Skull over the olfactory bulb was drilled. Injection depth was ~0.1 mm. Then, the virus was slowly injected at a rate of 2.3 nL every 6 seconds for 2 minutes (total 46 nL) using the nanoject II system and glass capillaries (#3-00-203-G/XL, Drummond). The concentration of the injected virus cocktail was $4 \times 10^9$ gc/mL for AAV2/1-FLEX-tTA2 and $1.25 \times 10^9$ gc/mL each for AAVDJ-TRE-XFP-WPRE. Before and after the injection, the needle was kept in place for 5 minutes. The hole on the skull was covered with superglue, and the exposed skull was then covered with dental cement. Post-surgery, the body of the mouse was kept warm to facilitate recovery. As toxic effects of prolonged XFP expression began to be observed after 4 weeks post injection, mice were sacrificed 3-4 weeks after virus injection[12].

### Sample preparation and tissue clearing with SeeDB2G
To obtain brain tissue, mice were intraperitoneally injected with an overdose of pentobarbital (P0776, TCI) in PBS to produce deep

anaesthesia, followed by intracardiac perfusion with a 25 mL PBS wash followed by 25 mL of 4% PFA in PBS. Excised brain samples were then fixed with 4% paraformaldehyde in PBS at 4 °C overnight. Brains were then embedded in 4% agarose (ThermoFisher, #16520-100). Cortical samples were cut into 120- or 320-μm-thick slices with a microslicer (PRO7N, Dosaka EM), and cleared with the SeeDB2G protocol[13]. Cleared cortical samples were then mounted in SeeDB2G (Omnipaque 350, Sankyo) on a glass slide using a 0.1 or 0.3 μm-thick silicone rubber sheet (AS ONE, #6-9085-12 or #6-9085-14, Togawa rubber) and glass coverslips (Marienfeld, No. 1.5H, #0109030091) as described previously[45].

To analyse long-rage axonal projections of M/T cells, a brain hemisphere was dissected, and the dorsal part and subcortical matter was trimmed away with forceps and a scalpel. The remaining part, containing all of the olfactory cortical areas, was flattened with a 1 mm spacer, and fixed with 4% PFA in PBS overnight at room temperature on a rocker[12]. Then, the sample was treated with ScaleCUBIC-1[46] (25% (wt/wt) urea (#219−00175, Wako), 25% (wt/wt) N,N,N',N'-tetrakis(2-hydroxypropyl)ethylenediamine (#T0781, TCI), and 15% (wt/wt) Triton X-100 (#12967−45, Nacalai-tesque) in H2O) for 24 hr to remove lipids from the lateral olfactory tract, washed with PBS, and then cleared with SeeDB2G as described previously[13,45].

## Confocal microscopy

Samples cleared with SeeDB2G were mounted on glass slides with 100 or 300 μm-thick silicone rubber spacer. Samples were imaged with an inverted confocal microscope, SP8 TCS (Leica) with HyD detectors. 20x (HC PL APO 20x/0.75 IMM CORR CS2) and 63x (HCX PL APO 63x/ 1.30 GLYC CORR 37 °C) objective lenses were used. Type G glycerol immersion (ThermoFisher, Cat#15336741) was used. XFPs were excited at 405, 488, or 552 nm lasers and emission signals were dispersed by a diffraction gating as described in legends to Fig. 1c. Pinhole size was set at 1 A.U. Images were acquired under "standard mode" and at 16-bit. Linearity was even worse under "photon counting mode". At both conditions, linearity was poor when recorded at 16-bit[18,19], and linearity correction was needed as described below.

## Linearity correction for fluorescence intensity acquired with Leica HyD detectors

**Measuring reference data and calculating the coefficients.** HEK293T cells were transfected with mNeonGreen, then following fixation they were imaged with an inverted confocal microscope with the laser power increasing for 0.2 to 3% with 0.2% intervals (Supplementary Fig. 1a, b, SP8 TCS with HyD detectors, Leica). Laser power was linearly controlled by ATOF. The intensity values for each pixel were then recorded, and the values from 0.2% to 1% were used to calculate a linear trend unique to each pixel (Supplementary Fig. 1c). This was then used to create a predicted value for each original intensity value which can then be used to create a fit to the equation below (Supplementary Fig. 1d).

$$y = ax + (b \times e^{cx}) - b \qquad (1)$$

In our microscope, $a = 0.9838$, $b = 1.1044$, and $c = 0.001$. As we can see in Supplementary Fig. 1b, values above 7000 were almost saturated and uncorrectable (Supplementary Fig 1d−f). We, therefore, discarded pixels above 7000 (assigned "not a number"). Pixels below 7000 were transformed to produce a corrected image. The MATLAB code for the linearity correction is available at Github (https://github.com/mleiwe/HyD_NonLinearCorrection).

**Evaluation of non-linearity corrections.** For successful linear unmixing, it is important to ensure that the ratios between channels are consistent across various intensity ranges. We, therefore, evaluated the consistency of the ratio (Ch4 / Ch4 + Ch5) using mNeonGreen. ROIs

for HEK293T cells were determined using Cellpose 2.0 (https://www.cellpose.org/)[47]. This was used to exclude background voxels from our ratio calculations. (Supplementary Fig 1g-i).

## Soma detection for brain samples

All neurite tracing and soma detection was performed in Nerurolucida 360 (MBF Biosciences, Version 2021.1.1, 2022.1.1, and 2023.1.1). For Soma detection, somas were initially detected with the automatic algorithm, followed by manual guided detection of the remaining soma.

## Manual neurite tracing for ground truth

Neurolucida 360 (MBF Biosciences, Version 2021.1.1, 2022.1.1, and 2023.1.1) was used for manual neurite tracing. For the creation of ground truth dendrite data (Fig. 5a), fully manual tracing was performed on the 63x images containing all of the channels. This enabled the accurate grouping of neurites into sets. For the manual tracing of 20x axon images (Supplementary Fig. 10c), user-guided tracing was performed (with directional kernels selected as the chosen method). The ground truth reconstruction data for Fig. 5 is archived at Zenodo (https://zenodo.org/records/11482026).

## Automatic detection of neurite fragments with Neurolucida 360

For fully automated tracing of neurites, each channel was individually normalized to its maximum, and then the maximum value was taken for each voxel to create a single channel volumetric image. This was then loaded into Neurolucida360 where firstly, the somata were auto detected, adjusting the parameters to ensure all soma were labelled (with remaining soma manually detected). Then, a two-stage auto-tracing procedure was used. First, a rayburst crawl was implemented to detect bright and large neurites, then a directional kernel algorithm was implemented to detect finer less bright neurites (seed density: dense; seed sensitivity: 80; refine filter: 2; trace sensitivity: 70; connect branches and remove traces shorter than… options: de-selected, for both tracing algorithms). Files were then saved as an xml file for export to our MATLAB analysis pipeline. To correct for errors where all Neurolucida algorithms selected the edge rather than centre of the neurite a custom-written MATLAB code was written to move the points to the centre of the neurite.

## Unmixing of overlapping fluorescence signals

Reference data was acquired with HEK293T cells expressing each of XFPs and cleared with SeeDB2G. mTagBFP2, mTurquoise2, and mAmetrine1.1 were excited at 405 nm and fluorescence at 410-468 nm (Ch1), 468-505 nm (Ch2), and 520−600 nm (Ch3) were acquired. mNeonGreen and YPet were excited at 488 nm and fluorescence at 495−525 nm (Ch4) and 525−555 nm (Ch5) were acquired. mRuby3 and tdKatushka2 were excited at 552 nm and fluorescence at 575-600 nm (Ch6) and 615−755 nm (Ch7) were acquired. Linearity correction was performed as described above. Based on the ratios across channels, linear unmixing was performed for sample images as described previously[16,17]. See Supplementary Fig. 2 for more details. See Supplementary Data 5 for the unmixing parameters used in this study. Images acquired with 20x and 63x objective lenses were processed with reference data with 20x and 63x, respectively. The MATLAB code for the linear unmixing is available at Github (https://github.com/mleiwe/LinearUnmixing). We also provide an ImageJ plugin for linear unmixing, Linear-unmixing-Qdye (https://github.com/daichimori/Linear-unmixing-Qdye).

## Post-hoc correction of chromatic aberration

Correction of chromatic aberration has been described previously[28]. MATLAB and Python codes are available at Github (https://github.com/mleiwe/ChromaticAberrationCorrection). Our program can correct non-uniform chromatic aberrations in cleared tissues *post hoc*.

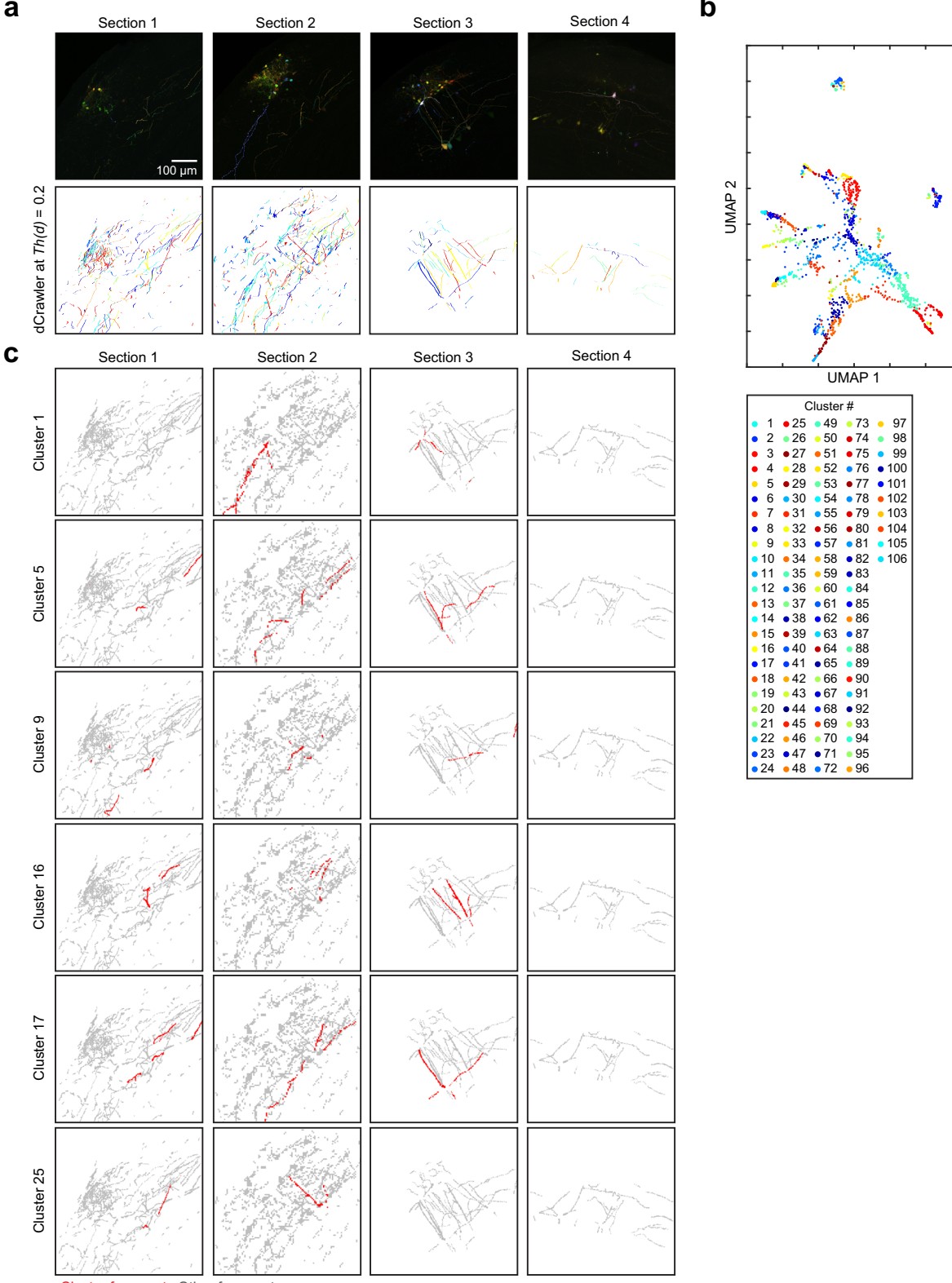

**Fig. 9 | Reconstruction of neurites across non-continuous samples. a**, 7-color Tetbow AAV (with CAG-tTA) was injected into the olfactory bulb of wild-type C57BL/6 N mice. Mitral/tufted cells and various types of periglomerular neurons are labelled. The mouse was 10-week-old female. Four consecutive slices (100 μm thick) were cut with vibratome, cleared with SeeDB2G, and imaged with confocal microscopy. Top row shows z-stacked images (volume = 581.82 × 581.82 × 109 μm) for each of the four brain slices. The bottom row reflects the traces extracted by Neurolucida 360, colour-coded to reflect their corresponding cluster, after running QDyeFinder at *Th(d)* = 0.2. **b** All fragments represented in UMAP-reduced colour space, colours represent the cluster it is assigned by QDyeFinder. **c** Representative clusters which clearly belong to single neurons that span several brain sections. This demonstrates that provided acquisition parameters are constant colour vector alone is sufficient for reconstruction, without using the physical continuity of neurites.

### Extracting colour vectors from neurite fragments

Fluorescence signals were extracted from the Neurolucida 360 traces using custom-built MATLAB codes (see also Supplementary Note 1 for Imaris). The putative fragments then went through a quality control pipeline (Supplementary Fig. 7) as follows. Firstly, traces were split into fragments at all branch points. Secondly, the brightness of each channel was evaluated by calculating the signal-to-noise ratio for the trace voxels compared to the background (non-traced) voxels. Each channel can be assessed manually or automatically, where a minimum signal-to-noise ratio of 3 was required for a channel to be included. Thirdly, a minimum length of a fragment was calculated (or manually inputted) to remove any short fragments that may have a large degree of colour variability. Fourthly, a minimum brightness was calculated (or manually inputted) in order to prevent analysis of any dim fragments which will have proportionally inaccurate vector normalised values. Finally, fragments where the colour changes significantly ($d > 0.3$) were split at the point of colour change. This often occurred when Neurolucida automatic tracing was performed as it does not use colour information for the tracing. Following these quality control steps, the mean raw intensity values of each fragment was obtained. Then, the median background of each channel was subtracted, and then normalized to the maximum per channel. The resulting values were then vector-normalized to obtain colour vectors.

### dCrawler

Clustering was performed from a matrix containing the vector normalised colour values, with the colour vector weighted by the magnitude for the centroid calculations. See Fig. 4 and Supplementary Note 2 for more details. The MATLAB and Python codes for dCrawler is available at Github (https://github.com/mleiwe/dCrawler).

### Calculation of optimum threshold distance

The optimal threshold distance (Th(d)) was calculated by using manually traced neurons from Neurolucida 360. The traces were processed the same as the automatically traced neurons. This created fragments that had a known neuron identity as well as a cluster identity. We determined that the correct cluster was the cluster that had the most fragments from the same neuron. This leads us to classify all fragments as true positive (correct cluster and correct neuron), false positive (correct cluster, different neuron), false negative (different cluster, correct neuron), and true negative (different cluster, different neuron). From there, an F1 score can be created for each individual neuron. These F1 scores were calculated for each neuron for a range of Th(d) from 0.05 to 1. The Th(d) with the highest median F1 score was considered optimum.

### Modelling and synthetic data

**Generation of synthetic data for Tetbow-labelled cells.** The generation of N-dimensional colour vectors for "cells" was an extension of the modelling performed in our previous study[12]. Briefly a Poisson probability distribution function was created for a range of copy numbers (average 0, 0.1, 0.2, 0.5, 1, 2, 4, 8 copies/colour/cell) to calculate the probability of up to 50 XFP copies being expressed per cell. This was then fitted to the number of "cells" specified for each dimension, with the order of the expression being shuffled for each channel to simulate the stochastic nature of transfection. The data was then vector normalised to place it into standard colour space (colour vectors).

**Percent discriminable.** This was performed as has been described in our previous study[12]. Discriminability (where $d > Th(d)$) was tested with 10,000 cells for between 1–7 XFPs.

**Percent unique.** To measure the number of "cells" that had a unique colour, we performed Monte-Carlo simulations (n = 200), by generating simulated colours in N dimensions, with a specified number of cells. "Cells" were classed as unique if they had no other cells within the threshold distance, Th(d). The percent of cells that were unique within each simulation was recorded and the mean and standard deviation were determined. All analysis was performed on MATLAB (v9.11) and are available on GitHub (https://github.com/mleiwe/QDyeFinder).

**Modelling with noise.** After modelling "cells", noise signals were introduced to recapitulate signal fluctuation in "neurites" (Supplementary Fig. 6). The noise followed normal distribution and SD per channel was changed. In this case, Th(d) was defined to cover 95% of the modelled "neurite" with noise. Percent discriminable was determined based on this Th(d) as described above. Modelling codes are available at GitHub (https://github.com/mleiwe/ModellingSuperTetbow).

### Quantitative comparison of traced data

We reconstructed neurites using Neurolucida and Imaris 10 Filament tracer (Imaris version 10.0). Because the format of the tracing data is different between the two platforms, we converted the data using the following procedure.

The neuron traces are processed using custom code written in Python3.9.10. The file types included are Neurolucida XML and Imaris filaments, which are converted to the standard SWC format. The manual tracing was performed in Neurolucida360.

First, the Imaris 10 Filament Tracer was used to obtain the automatic tracing result. Subsequently, the filaments were converted directly from Imaris to 'SWC' morphologies using a custom-written SWC exporter, which is then parsed in Python. The code is available on GitHub (https://github.com/Elsword016/Swc-plugins-for-Imaris-10). For Neurolucida XML, we converted it to a MATLAB struct using codes that were already integrated into the main QDyeFinder algorithm. The MATLAB struct is then parsed in Python using the mat73 library.

For our quantification metric, we calculated "% reconstructed", which indicates how much of the ground truth fragments were covered by QDyeFinder or Imaris (area of intersection / area of ground truth). We increased the width of the traces of QDyeFynder or Imaris by 10 pixels to negate small misalignments of the traces (<10 pixels).

### Statistics and reproducibility

Statistical analyses used are explained in figure legends. Sample sizes were not pre-determined. We did not perform blind analyses. Instead, we analysed all the possible neurites in the images to exclude any human biases. Numbers of synthetic "neurons" (10,000 cells) in simulation studies were determined based on the computational costs. Representative images and their data in Figures were chosen from at least three independent samples with similar conclusions. Labelling densities with Tetbow varied between experiments; we have excluded samples where the labelled neurons were too sparse (<10) or too dense (>500). Exclusion criteria for QDyeFinder pipeline are explained in Fig. 2.

### Reporting summary

Further information on research design is available in the Nature Portfolio Reporting Summary linked to this article.

## Data availability

All of the image data acquired in this study has been deposited to SSBD:repository (https://ssbd.riken.jp/repository/346/, https://doi.org/10.24631/ssbd.repos.2024.05.346). Neuronal reconstruction data made with Neurolucida 360 and Imaris 10 Filament Tracer are available at Zenodo (https://doi.org/10.5281/zenodo.11482026) and Neuromorpho.org (https://neuromorpho.org/index.jsp). At Neuromorpho.org, data are associated with "Imai" archive and Pubmed ID of this paper. Source data are provided with this paper.

## Code availability

Modelling codes are available at GitHub (https://github.com/mleiwe/ModellingSuperTetbow, https://doi.org/10.5281/zenodo.11233271). *Post-hoc* chromatic aberration correction codes (both MATLAB and Python) are available at GitHub (https://github.com/mleiwe/ChromaticAberrationCorrection, https://doi.org/10.5281/zenodo.11180569, 2024). Linearity correction program for HyD detectors is available at GitHub (https://github.com/mleiwe/HyD_NonLinearCorrection, https://doi.org/10.5281/zenodo.11180581, 2024). Linear unmixing code for MATLAB is available at GitHub (https://github.com/mleiwe/LinearUnmixing, https://doi.org/10.5281/zenodo.11180583, 2024). ImageJ plugin for linear unmixing, Linear-unmixing-Qdye, is available at GitHub (https://github.com/daichimori/Linear-unmixing-Qdye, https://doi.org/10.5281/zenodo.11232721). dCrawler written in MATLAB and Python is available at GitHub (https://github.com/mleiwe/dCrawler, https://doi.org/10.5281/zenodo.11180589). QDyeFinder written in MATLAB as well as test data is available in GitHub (https://github.com/mleiwe/QDyeFinder, https://doi.org/10.5281/zenodo.11180585). SWC exporter for Imaris Filament Tracer data written in Phython is available at GitHub (https://github.com/Elsword016/Swc-plugins-for-Imaris-10, https://doi.org/10.5281/zenodo.11232963). Additional resources and protocols are available in our GitHub (https://github.com/TakeshiImaiLab) and SeeDB Resources (https://sites.google.com/site/seedbresources/). Requests for additional data should be directed to and will be fulfilled upon request by the Lead Contact, Takeshi Imai (imai.takeshi.457@m.kyushu-u.ac.jp).

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

## Acknowledgements

We thank M. Nishihara, T. Ohmine, and E. Nozoe for technical assistance; Elvire Guiot for sharing their application note on Leica HyD; Hisagi for sharing Tokyo metro map in Fig. 1a (https://commons.wikimedia.org/wiki/File:Tokyo_metro_map.png); The MathWorks, Inc. for permission to use a test image in Supplementary Fig. 14c. We also appreciate the technical assistance from The Research Support Center, Research Center for Human Disease Modeling, Kyushu University Graduate School of Medical Sciences, which is partially supported by the Mitsuaki Shiraishi Fund for Basic Medical Research. This work was supported by Brain/MINDS project (JP20dm0207055 to TI) and iBrain/MINDS project (JP23wm0525012 to TI) from AMED, JST CREST program (JPMJCR2021 to TI), Grants-in-Aids from MEXT (JP16H06456, JP21H00205, JP21H05696, JP24H02308, and JP24H02312 to TI), JSPS KAKENHI (JP17H06261 and JP21K19355 to TI; JP19K16261 and JP21K06411 to MNL; JP19K06886 and JP24K02132 to SF; JP21H02140 and JP22K18373 to S.I.; JP20J23361 to D.M.), the JST SPRING Grant JPMJSP2136 (to D.M.), a grant from the Uehara Memorial Foundation (to TI), a grant from the Sumitomo Foundation (to S.F.), a grant from the Ichiro Kanahara foundation (to S.F.), a grant from Daiichi Sankyo Foundation of Life Science (to S.F.), and a grant from Brain Science Foundation (to S.F.). D.M. was a predoctoral research fellow (DC1) of JSPS. B.S. is a MEXT scholar for international students.

## Author contributions

M.N.L. performed most of the data analysis. S.F. and T.B. performed sample preparation and super-multicolour imaging of cultured cells and brain samples. D.M. and M.N.L developed linearity correction program. D.M. also developed ImageJ plugin for linear unmixing. B.S. wrote Python versions of code for a post-hoc chromatic aberration program and dCrawler. B.S. also wrote swc exporter code for Imaris Filament Tracer data and performed benchmarking of tracing software. R.S. and S.I. assisted initial phase of the study. T.I. supervised the project. M.N.L, S.F, T.B, and T.I. wrote the manuscript with inputs from all the authors.

## Competing interests

The authors declare the following competing interests: T.I., M.N.L., and S.F. have filed a patent application (JP-A-2023-17644) related to QDye-Finder. The remaining authors declare no competing interests.
