## [Peer Review File · Nature Communications]

Reviewers' Comments:

Reviewer #1:

Remarks to the Author:

In this manuscript, Leiwe et al. presented a strategy for the reconstruction of neuronal morphology based on their 7 FP labeling method and automated reconstruction pipeline. To increase the discernibility of the neurons using color information, the authors screened and employed a set of 7 FPs that are optimal (emission spectra overlapping, brightness, color uniformity, trafficking, etc). Spectral leakage that comes from the overlapping emission spectra is removed via linear unmixing to acquire "clean" 7-color images. Thereafter, the images go through an automated pipeline. First, over-segmentation (using commercially available software) is performed to get super-pixels that contain neurites, and the color for each super-pixel is evaluated. Then, those super-pixels are merged using their color information to reconstruct the neurons.

While I do appreciate the massive scale of this work that ranges from screening fluorophores to mesoscale reconstruction, I also have to point out that each element presented in this manuscript was demonstrated by others years ago. Furthermore, the elements are not quantitatively compared to the existing methods at all and the manuscript does not seem to contain new striking results. What follows are specific concerns about the manuscript and the results.

Major concerns.

a. Employing more than 3 FPs to increase the discernibility of neurons has been demonstrated multiple times. In 2019, Veling et al. demonstrated 5-FP imaging. Admittedly, the number of colors in this work was not as many as Leiwe et al., but it should be noted that they devised an advanced form, or a "digital" version, of BrainBow to ensure color discernibility. In Li et al., the authors generated transgenic Bitbow *Drosophila* lines, established the experimental protocol, imaged the entire nerve system, and developed a computational pipeline for neuronal morphology reconstruction. The algorithmic pipeline presented in the manuscript is very similar to Sümbül et al. which was published 6 years ago. In their pipeline for neuronal morphology reconstruction, the image is denoised to deal with the color non-uniformity and then over-segmented, and the segments are merged through spectral clustering. Considering such high-level similarity to the earlier works, the reviewer thinks that the authors need to justify their method through proper comparison.

b. (please note that this comment is related to the minor comment d) The authors made a bold claim that 99.9% of the neuronal pairs can be differentiated with 7 FPs as if 7 were a magic number. However, this is based on a strong assumption that (1) color is uniform within each neuron (which contradicts to authors' following statement "the distribution of XFPs were not completely uniform at a local level (< several microns), especially in thin axons"), (2) image is noise-free (which also contradicts to authors' following statement "Shot noise contributed to significant fluctuations"), and (3) the color of each neuron is a random variable that is drawn from a uniform distribution. This means seven is not a magic number – and this makes the increase in color number only incremental improvement over earlier works.

Minor comments.

a. The computational pipeline proposed in this work is full of heuristics:

"We need to discard the channel when S/N in this range was <2.5",

"We also needed to discard very short neurite fragments",

"We found 186 that $d > 0.1$ when the lengths of sub-fragments become $<5 \mu\text{m}$ ",

"We also discarded ROIs with insufficient brightness",

"If $d > 0.2$, we defined it as an inaccurate sub-fragment",

"To determine a magnitude threshold, we 193 found the smallest magnitude cut-off where less than 5% of the sub-fragments were 194 inaccurate",

"We, therefore, broke down the fragments further 200 when the colour vectors in adjacent sub-fragments (segmenting by the minimum fragment 201 length described above) of the fragment are separated by $d > 0.3$ ",

and the list goes on and on. I understand that processing a large dataset often involves manual

steps that are often not discussed in published papers, but such a heuristic pipeline hardly has any intellectual value or general usage.

b. The authors pointed out that the k-means algorithm requires the number of clusters. This is true, but it is not fundamentally different from setting the threshold in the clustering algorithm. It is just that some unsupervised clustering algorithms take the number of clusters as the hyperparameter (e.g., k-means) and some others take the threshold value (e.g., DBSCAN). In addition, there are multiple ways to estimate the number of neurons (e.g., Smbl et al. showed that the number of neurons can be estimated from a Dirichlet process).

c. I believe there is enough consensus that having more colors helps discern a large number of objects, and hence Fig. 1a seems unnecessary.

d. Regarding Fig. 2e, isn't the whole point of employing 7 colors to disambiguate thin neurites (but not soma)? Detecting and disambiguating soma, with reasonable image quality, is nearly trivial with even grayscale images.

1. Veling et al., Identification of Neuronal Lineages in the Drosophila Peripheral Nervous System with a "Digital" Multi-spectral Lineage Tracing System, Cell Reports (2019)

2. Li et al., Bitbow Enables Highly Efficient Neuronal Lineage Tracing and Morphology Reconstruction in Single Drosophila Brains, Front. Neural Circuits (2021)

3. Smbl et al., Automated scalable segmentation of neurons from multispectral images, NeurIPS (2016)

Reviewer #2:

Remarks to the Author:

This work describes an improved 7-color fluorescent labeling strategy and a new computational package to trace and reconstruct the morphology of neurites for brain connectome (projectome) mapping, which is a great technology advance compared with previously reported Brainbow and Tetbow. While the work is of significance, I have a few major concerns:

1. The accuracy of color separation for combinatorial labeling

a. The authors have conducted single-color XFP labeling in HEK293 cells to benchmark color separation, but only performed computational simulation for 2-color XFP labeling or more, which is insufficient since most of the neurons will be labeled by more than 1 color in vivo. I wonder if the author could repeat the HEK293 experiments but with dual color labeling (in total 7X6 combinations) to experimentally measure the accuracy of the linear unmixing of 2-color XFP labeling using QDyeFinder.

b. A major issue with Brainbow-type of approach is potential color hue drifts between soma and neurites. Since the new XFP are introduced for tracing in this paper, the author should prove and comment on whether the distribution of XFPs is even in soma versus neurites.

c. In addition, potential chromatic aberration across different z-plane in thick tissue can also impact the fidelity of tracing. Could the author describe how they evaluated chromatic aberration across z?

2. Cross-method comparison with other color separation and tracing methods

a. The author has included Table 1 for a comparison of different tracing methods, yet there lack of quantitative analysis. Can the author test previously published methods and compare their performance with dCrawler?

b. The author demonstrated that QDyeFinder is a conceptually new method. Yet, there lacks of comparison with existing methods, why is QDyeFinder conceptually novel? Can the author include a comparison table similar to dCrawler and benchmark the performance of QDyeFinder with existing approaches?

3. Applicability of the method

a. Is this method applicable to other microscopes beyond the Leica SP8 microscope?

b. Since there are a few existing whole-brain tracing approaches, besides tracing within the same 100- and 300-um tissue block, does the method work across different tissue blocks?

c. Based on the # of distinguishable color hues, is there an upper limit of the # of cells that can be traced in a single experiment?

d. Can the image be registered to Allen Common Coordinate Framework (CCF) to allow cross-animal and cross-lab integration and comparison?

4. Biological insights into the morphological reconstruction

The authors comment at the discussion session of the paper that, in comparison with MAPseq, Tetbow-based approaches are better suited to morphological studies. However, MAPseq with the recent capabilities of cell-typing using in situ sequencing can reveal basic connectome principles using cell-type and single-cell resolved projectome mapping. Can the author demonstrate and comment on the major biological insights that morphological reconstruction can reveal while missing cell-type information?

Reviewer #3:

Remarks to the Author:

The methods proposed on this manuscript aims on solving two important constrains on neuronal reconstruction: (1) automation and (2) dense reconstructions within the same tissue. As an approach they combine stochastic labeling of neurons with 7 fluorescent markers and a computational strategy to cluster and identify the labeling in different neurons. On strength of this approach is that the physical continuity of the neurite fragments is less important as the combination of fluorescent proteins serves as a barcode to support the reconstruction.

The idea is very interesting, and the authors test its feasibility on dendrites and on the inter-areal portion of axons, which are usually thicker. I was surprised for not seeing a figure dedicated to the reconstruction of the thinner and more local arbors of the axons, and I think that would increase the impact and adoption of this approach.

My major recommendation is for the authors to either add local arbor axonal reconstructions to the neurons showed in Figures 6 and 7, or discuss what are limitations of the approach that at this point prevents them from obtaining the reconstructions of the fine collaterals. These fine collaterals are important because those are usually the locations of synapses.

Minor points

1. The authors present estimations of percentage of neurons with unique labels for different total numbers of labeled neurons. I would suggest they discuss where do they see the sweet spot for this approach in terms of number of labeled neurons versus quality of reconstructions. When does multiple neurons labeled with the same combination of proteins becomes problematic given the quality metrics for soma, dendrite and axonal reconstructions?

2. Related with the previous point the authors mention: "As many of neurons are uniquely labelled by the combinatorial expression of 7 XFPs, we considered that we should be able to identify neurites for different neurons solely based on the colour hue information". How many times do the automatic reconstructions lead to reconstructions with 2 somas? This could potentially be a metric to predict the percentage of "non-unique labels" that would be tolerated.

3. Page 6: " We, therefore, decided to extract colour vector data for regions of interest (ROIs) consisting of multiple pixels". How many pixels is the minimum?

4. The authors could discuss or compare the size of their reconstructions with the state of the art reconstructions of the same cell types. E.g. State of the art reconstructions of cell type "X" (citation) achieve "Y" length of axon and "Z" branch points, our algorithm finds "A" length and "B" branchpoints per neuron of cell type "X" and in addition can reconstruct a total of "C" neurons within the same sample.

5. The first author of reference 4 is Microns Consortium and not Bae.

Summary of Revision

We thank all the reviewers for their critical and constructive comments. Here is the summary of the revision.

Supplementary Fig. 3. We now show that linear unmixing is useful for epifluorescence images taken with conventional filter/mirror sets.

Supplementary Fig. 4. We show that single-coloured and dual-coloured cells can be distinguished.

Supplementary Fig. 6. We performed a simulation of neurite fragments with noise, assuming more realistic situations. We found that 7-colour images are still much more advantageous than conventional 3-colour images.

Supplementary Fig. 12. We demonstrated auto-tracing of fine axonal collaterals using a 63x objective lens.

Supplementary Fig. 14. Comparison with the current state-of-the-art (Imaris 10 Filament Tracer) was performed. Imaris 10 was released after the submission of this manuscript, and its auto-tracing performance is much better than previous versions incorporating machine learning procedures. We show that Imaris still falls short of the performance of manual tracing and QDyeFinder.

Supplementary Fig. 15. Multiple popular clustering algorithms were compared with dCrawler in our QDyeFinder pipeline. We now show that dCrawler outperforms k-means clustering previously used in Sümbül et al.

Supplementary Table 1. Comparison with other tracing methods is summarized.

Supplementary Note 1. In addition to a common .swc format, we now support another format (.hoc) used in Imaris because Imaris is also a common software.

We have modified the description in the Results section accordingly. In addition, we added a discussion related to optimal parameters and imaging conditions in response to reviewers' suggestions. We believe that a comparison with previous methods (particularly Sümbül et al. and Imaris 10 Filament Tracer) was adequately performed in this revision. We believe that all the reviewers will be satisfied with this revision.

In the rebuttal letter, the reviewers' comments are *italicised*, and our responses are provided for each comment. New changes in this revision are **highlighted** in both the manuscript and the rebuttal letter.

Reviewer #1 (Remarks to the Author):

In this manuscript, Leiwe et al. presented a strategy for the reconstruction of neuronal morphology based on their 7 FP labeling method and automated reconstruction pipeline. To increase the discernibility of the neurons using color information, the authors screened and employed a set of 7 FPs that are optimal (emission spectra overlapping, brightness, color uniformity, trafficking, etc). Spectral leakage that comes from the overlapping emission spectra is removed via linear unmixing to acquire “clean” 7-color images. Thereafter, the images go through an automated pipeline. First, over-segmentation (using commercially available software) is performed to get super-pixels that contain neurites, and the color for each super-pixel is evaluated. Then, those super-pixels are merged using their color information to reconstruct the neurons. While I do appreciate the massive scale of this work that ranges from screening fluorophores to mesoscale reconstruction, I also have to point out that each element presented in this manuscript was demonstrated by others years ago. Furthermore, the elements are not quantitatively compared to the existing methods at all and the manuscript does not seem to contain new striking results. What follows are specific concerns about the manuscript and the results.

Major concerns.

a. Employing more than 3 FPs to increase the discernibility of neurons has been demonstrated multiple times. In 2019, Veling et al. demonstrated 5-FP imaging. Admittedly, the number of colors in this work was not as many as Leiwe et al., but it should be noted that they devised an advanced form, or a “digital” version, of BrainBow to ensure color discernibility. In Li et al., the authors generated transgenic Bitbow Drosophila lines, established the experimental protocol, imaged the entire nerve system, and developed a computational pipeline for neuronal morphology reconstruction. The algorithmic pipeline presented in the manuscript is very similar to Sümbül et al. which was published 6 years ago. In their pipeline for neuronal morphology reconstruction, the image is denoised to deal with the color non-uniformity and then over-segmented, and the segments are merged through spectral clustering. Considering such high-level similarity to the earlier works, the reviewer thinks that the authors need to justify their method through proper comparison.

We agree that it is important to compare our new analysis pipeline with prior art. Veling et al. (nBitbow) indeed utilized binary expression of 5 XFPs; however, they did not perform any automated analysis. Only Sümbül et al. tried automated analysis of the Brainbow data. We apologize for our oversight of the Sümbül et al., which is now discussed in the main text (Line 386-393). Unfortunately, the program code made by Sümbül et al. was not publicly available, precluding detailed evaluation of their original code (especially the earlier part for superpixelisation). Nonetheless, we added a comparison with Sümbül et al. as much as possible, namely k-means clustering vs. dCrawler (Supplementary Fig. 13).

As the reviewer points out, the earlier steps in the analysis pipeline are conceptually similar between Sümbül et al. and our current study. The major differences are 1) the use of 7 XFP instead of 3 XFP in Sümbül et al., and 2) the use of dCrawler instead of K-means clustering in Sümbül et al. The combination of these two parts has dramatically improved the discriminability of neurons and achieved the identification of “a single neuron” from densely labelled (~100) neurons, which has never been achieved in the previous method (Sümbül et al.). We, therefore, believe that our strategy, while based on multiple minor improvements, achieved a major step forward in terms of performance.

b. (please note that this comment is related to the minor comment d) The authors made a bold claim that 99.9% of the neuronal pairs can be differentiated with 7 FPs as if 7 were a magic number. However, this is based on a strong assumption that (1) color is uniform within each neuron (which contradicts to authors' following statement “the distribution of XFPs were not completely uniform at a local level (< several microns), especially in thin axons”), (2) image is noise-free (which also contradicts to authors' following statement “Shot noise contributed to significant fluctuations”), and (3) the color of each neuron is a random variable that is drawn from a uniform distribution. This means seven is not a magic number – and this makes the increase in color number only incremental improvement over earlier works.

We thank this reviewer for raising this important issue. Our old Fig. 2b demonstrated that 7 XFPs dramatically improve the discriminability compared to 3 XFPs when soma data (with zero noise) was used and “a defined threshold value” was used. However, as pointed out by this reviewer [points (1) and (2)], it is not obvious whether the conclusion remains the same when samples contain significant levels of noise signals.

We totally agree that it may not be fair to use the same threshold value, $Th(d)$, to compare the discriminability in 3 XFPs vs. 7 XFPs data.

Fig. R1. Modeling colours in neurite fragments with noise (Supplementary Fig. 6).

To address this issue, we examined how the noise level impacts the appropriate $Th(d)$ and discriminability based on the $Th(d)$ (Fig. R1a). We considered that the fluorescence intensity value in each channel (Ch) contains a random amount of noise, which follows the normal distribution. We changed the noise size and determined the $Th(d)$ that contains 95% of the data points with noise. Then, we used that $Th(d)$ to calculate the discriminability.

We used modelled (synthetic) cells as in Fig. 2a, b and generated modelled “neurite fragments” with noise to determine optimum $Th(d)$ (Fig. R1b). Based on the $Th(d)$, we calculated percent discriminability as in Fig. 2d (Fig. R1c). As expected, the $Th(d)$, which contains 95% of the data points in this case, became larger as the number of XFPs increase. For example, when standard deviation (SD) of the noise was 0.1 per channel, $Th(d)$ was 0.239 with 3 XFPs and 0.335 with 7 XFPs. Nevertheless, we found that when the SD of the noise is up to 0.1 (which is realistic based on old Supplementary Fig. 5), discriminability was much higher with 7 XFPs than with 3 XFPs (Fig. R1d). For example, when 7 XFPs were expressed at 2 copies per cell per colour and SD of the noise was 0.1, neurons were 98.5% discriminable, which was much greater than 3 XFPs (only 91.2% discriminable). We also found that discriminability was highest (99.9% discriminable with 7 XFPs) when 1 copy/cell/colour was expressed.

In summary, we conclude that 7 XFPs is much better than 3 XFPs, even if the signals in the neurite fragments contain a certain amount of noise ($SD < 0.1$).

The reviewer’s point (3) has already been considered in the model. We do not think that 7 is the magic number. The more is better, but the number is simply limited by the available fluorescent proteins.

This new analysis (Fig. R1) is now in Supplementary Fig. 6 and is described in the Result section (Line 134-141).

Minor comments.

a. The computational pipeline proposed in this work is full of heuristics:

"We need to discard the channel when S/N in this range was < 2.5 ",

"We also needed to discard very short neurite fragments",

"We found 186 that $d > 0.1$ when the lengths of sub-fragments become $< 5 \mu\text{m}$ ",

"We also discarded ROIs with insufficient brightness",

"If $d > 0.2$, we defined it as an inaccurate sub-fragment",

"To determine a magnitude threshold, we 193 found the smallest magnitude cut-off where less than 5% of the sub-fragments were 194 inaccurate",

"We, therefore, broke down the fragments further 200 when the colour vectors in adjacent sub-fragments (segmenting by the minimum fragment 201 length described

above) of the fragment are separated by $d > 0.3$ ", and the list goes on and on. I understand that processing a large dataset often involves manual steps that are often not discussed in published papers, but such a heuristic pipeline hardly has any intellectual value or general usage.

We agree that many of the methods used in biology are full of heuristics and have little intellectual value. However, no clever idea works in the real world without heuristics. This is reality, especially in biology.

To be clear, Sümbül et al. only demonstrated successful separation in synthetic simulated data. They failed to isolate neurites from just one neuron in actual neuron data. Anyone can easily come up with the same idea, but it won't work without optimizing plenty of parameters based on actual biological data. In the past several years, we have invested a tremendous amount of effort to improve both wet (FP choice, expression level, # of FPs, clearing methods) and dry analyses (pixel-based vs. superpixel-based, cut-off values, raw intensity vs. vector-normalized value, k-means vs dCrawler, etc.). This method won't work without the combination of these heuristics. Once successfully applied to biological samples, it can be further improved for general usage, as you see in the history of various biological methods (i.e., deep sequencers, scRNA-seq, iPS cells, super-resolution microscopy, etc.). Theoretically possible and practically possible are hugely different in biology and engineering fields.

b. The authors pointed out that the k-means algorithm requires the number of clusters. This is true, but it is not fundamentally different from setting the threshold in the clustering algorithm. It is just that some unsupervised clustering algorithms take the number of clusters as the hyperparameter (e.g., k-means) and some others take the threshold value (e.g., DBSCAN). In addition, there are multiple ways to estimate the number of neurons (e.g., Sümbül et al. showed that the number of neurons can be estimated from a Dirichlet process).

Indeed, different clustering methods require different parameters. There are three issues to discuss.

1) The first point is which parameters can be "easily" obtained from the image data. It is true that there are ways to estimate the number of clusters in k-means; however, in that case, we have to determine the number of clusters in each of the image data. In contrast,

in dCrawler, we can use the same $Th(d)$ once we establish the labelling methods. In multiple samples we used, the optimum $Th(d)$ was ~ 0.2 , meaning that we may not need to care about this parameter as long as we used 7-colour Tetbow. Pros and cons are summarized in Table 1.

2) Different clustering methods assume different distributions of the data points in each cluster. It is, therefore, important to choose a method that assumes the distribution in actual biological samples. As summarized in Table 1, dCrawler seems to be better than Mean Shift Clustering and DBSCAN in that sense. DBSCAN is dangerous, as multiple clusters can be easily fused. The difference between different clustering methods is now summarized in the legend in Table 1.

3) Performance was now quantitatively evaluated using actual biological data (Fig. R2; Also in Supplementary Fig. 13). Here, we compared the performance of the four clustering methods with the actual data in Fig. 5. We found that k-means is the worst, even when we used the best parameter ($k = 42$).

Based on the above three points, we concluded that QDyeFinder utilising dCrawler performs the best. This is now mentioned in the main text (Line 386-393).

Fig. R2. Algorithm comparison in QDyeFinder (Supplementary Fig. 13). Fig. 5 data with a ground truth was used to evaluate the performance of the clustering.

c. I believe there is enough consensus that having more colors helps discern a large number of objects, and hence Fig. 1a seems unnecessary.

We agree, but the question is how much it improves using 7 XFPs. It should be helpful for the audience to know how much improvement we can expect with 7-colours with quantitative metrics. If necessary, we can omit Fig. 1a.

d. Regarding Fig. 2e, isn't the whole point of employing 7 colors to disambiguate thin neurites (but not soma)? Detecting and disambiguating soma, with reasonable image quality, is nearly trivial with even grayscale images.

Fig. 2e is aimed to evaluate the possible number of colour combinations produced by 3- vs. 7-colour Tetbow. This analysis is just to support the conclusion of the analysis based on the synthetic data (Fig. 2a-d), not aimed at identifying the soma.

Figs. 3-7 employ 7 colours to disambiguate thin neurites.

We thank this reviewer for raising critical issues related to Sümbül et al., which used 3-color Brainbow and k-means clustering. In this revision, we added more detailed comparisons between 3 XFPs vs. 7 XFPs and k-means vs. dCrawler. We hope this reviewer agrees that our current study represents a major advance over the prior art in terms of its performance.

Reviewer #2 (Remarks to the Author):

This work describes an improved 7-color fluorescent labeling strategy and a new computational package to trace and reconstruct the morphology of neurites for brain connectome (projectome) mapping, which is a great technology advance compared with previously reported Brainbow and Tetbow. While the work is of significance, I have a few major concerns:

1. The accuracy of color separation for combinatorial labeling

a. The authors have conducted single-color XFP labeling in HEK293 cells to benchmark color separation, but only performed computational simulation for 2-color XFP labeling or more, which is insufficient since most of the neurons will be labeled by more than 1 color in vivo. I wonder if the author could repeat the HEK293 experiments but with dual color labeling (in total 7X6 combinations) to experimentally measure the accuracy of the linear unmixing of 2-color XFP labeling using QDyeFinder.

According to this suggestion, we evaluated single-colour vs. dual-colour labelling. However, we do not need to test 42 combinations, as a single laser can excite only 2 or 3 XFPs at a time. We compared single-colour vs. dual-colour labelling for mTagBFP2, mTurquoise2, and mAmetrine1.1 with 405 nm; mNeonGreen and YPet with 488 nm; mRuby3 and tdKatushka2 with 552 nm.

In some combinations, fluorescence intensities were not 1:1 when equal amounts of plasmids were transfected. However, this is not a problem because the signal intensities will be normalised for each channel at later stages. XFPs were identified in an all-or-nothing fashion for both single-coloured and dual-coloured samples (Fig. R3; Also in Supplementary Fig. 4). These results indicate that multiple XFPs are independently measured from our Tetbow samples after the linear unmixing.

Fig. R3. Distinguishing between single-coloured vs. dual-coloured cells (Supplementary Fig. 4).

b. A major issue with Brainbow-type of approach is potential color hue drifts between soma and neurites. Since the new XFP are introduced for tracing in this paper, the author should prove and comment on whether the distribution of XFPs is even in soma versus neurites.

Admittedly, the colour hues at soma, dendrites, and thin axons are not necessarily consistent. This is most likely due to localization bias of XFPs and different signal/background ratios under different brightness. Therefore, dendrites and axons should be analysed separately. This limitation is now mentioned in the discussion (Line 375-378).

c. In addition, potential chromatic aberration across different z-plane in thick tissue can also impact the fidelity of tracing. Could the author describe how they evaluated chromatic aberration across z?

It is absolutely true. In that case, different colours will appear at different z positions for multicolour-labelled neurites. This is now mentioned in Line 174-177. More detailed procedures for the evaluation and correction of chromatic aberrations are described in Ref. 26 (Leiwe et al., 2021).

2. Cross-method comparison with other color separation and tracing methods

a. The author has included Table 1 for a comparison of different tracing methods, yet there lack of quantitative analysis. Can the author test previously published methods and compare their performance with dCrawler?

See our response to Reviewer #1 (minor-b, See also Fig. R2 and Supplementary Fig. 13)

b. The author demonstrated that QDyeFinder is a conceptually new method. Yet, there lacks of comparison with existing methods, why is QDyeFinder conceptually novel? Can the author include a comparison table similar to dCrawler and benchmark the performance of QDyeFinder with existing approaches?

In our hand, Imaris 10 with Filament Tracer, released in 2023, performs the best among the fully automated neurite reconstruction software. The machine learning process much improved the performance with Imaris 10. However, it is difficult to benchmark the

performance quantitatively because 1) the performance depends heavily on the type of images (sparse vs. dense labelling) and training dataset for the machine learning process, and 2) the data format used for Imaris is very different from ours. We, therefore, show a typical example of tracing errors with Imaris in Fig. R4 (also in Supplementary Fig. 12). Side-by-side comparison with ground truth and QDyeFinder is shown. I believe that Fig. R4 shows the obvious differences between the current state of the art (Imaris 10) and our strategy.

Fig. R4. Imaris vs QDyeFinder (Supplementary Fig. 13).

We also prepared a new table showing the pros and cons of current and existing strategies for neurite reconstruction (Supplementary Table 1). Unique advantages are now highlighted.

3. Applicability of the method

a. Is this method applicable to other microscopes beyond the Leica SP8 microscope?

Yes. We can perform unmixing for multicolour fluorescence images taken with conventional filter/mirror units with both epifluorescence and confocal microscopy. Fig. R5 and Supplementary Fig. 3 show unmixing of epifluorescence images using conventional filter/mirror units.

Fig. R5. Linear unmixing of fluorescence images taken with conventional filter/mirror units (Supplementary Fig. 3).

It is advised to use an objective lens optimized for SeeDB2 to minimize chromatic aberrations. However, even with a non-optimal lens, chromatic aberrations can be corrected according to our previous study (Ref. 26, Leiwe et al., 2021).

b. Since there are a few existing whole-brain tracing approaches, besides tracing within the same 100- and 300-um tissue block, does the method work across different tissue blocks?

Yes. This is exactly one of the unique advantages of our new method over the existing methods: We do not rely on the physical continuity of the neurites. This is important because high-resolution imaging requires a high NA objective lens, which typically has a short working distance (WD) and requires brain slicing. To highlight this important advantage, we have moved the old Supplementary Fig. 9 to the main Fig. 9, showing the tracing across multiple sections. This point is also described in the last part of the Result section.

c. Based on the # of distinguishable color hues, is there an upper limit of the # of cells that can be traced in a single experiment?

As we use the colour hues to distinguish neurites, the upper limit of the # of cells is

equal to the number of distinguishable colour hues.

d. Can the image be registered to Allen Common Coordinate Framework (CCF) to allow cross-animal and cross-lab integration and comparison?

There are several methods to transform the volume images to CCF. In general, the original data should be close to the whole-brain scale for this purpose. However, we have not tried to analyse images at the whole-brain scale, simply because we do not have access to such a microscope. It will take at least months to take 7-colour images for the whole brain if we use a conventional point-scanning confocal microscope. The data size will be at least ~100TB. Typical whole-brain imaging methods (e.g., light-sheet microscopy) take images at much lower resolution to accommodate the data size below ~TB. This topic is interesting but far beyond the scope of this study.

4. Biological insights into the morphological reconstruction

The authors comment at the discussion session of the paper that, in comparison with MAPseq, Tetbow-based approaches are better suited to morphological studies. However, MAPseq with the recent capabilities of cell-typing using in situ sequencing can reveal basic connectome principles using cell-type and single-cell resolved projectome mapping. Can the author demonstrate and comment on the major biological insights that morphological reconstruction can reveal while missing cell-type information?

Morphological information is obviously important information, and this is why a lot of neuroscientists are spending a lot of effort on projectome and connectome analyses. Morphological information should tell a lot about how neurons receive synaptic inputs, integrate them, and send the output to other neurons. This is now mentioned in the Discussion section (Line 358-363). Our Tetbow approach may also be useful to show diversity within the specific cell type when combined with a cell type-specific Cre driver. Fig. 7 is an example where Pcdh21-Cre was used to express Tetbow AAVs specifically in mitral cells.

As for the biological insights, we can show two examples from our previous studies. Fig. R5 (adopted from Ke et al., Nat Neurosci 2013) shows the morphology of lateral dendrites in multiple mitral cells. They are all mitral cells, but there seems to be functional subclasses: One group has several dendrites, and the other group has only a

few dendrites. While this tracing was performed by manual tracing, Tetbow and QDyeFinder should much facilitate the tracing process. Fig. R6 (adopted from Fujimoto et al., *Dev Cell* 2023) shows the morphology of mitral cells in the developmental stage. The morphology clearly tells that the receptive field of mitral cells are gradually tuned to from multiple glomeruli (ORs) to just one glomerulus (OR). Tetbow and QDyeFinder will facilitate morphological analyses.

Fig. R5. Morphology of lateral dendrites for mitral cells connecting to the same glomerulus. Lateral dendrites were manually traced after clearing with SeeDB. Adopted from Supplementary Figure 13 of Ke et al., *Nat Neurosci* 2013.

Fig. R6. Morphology of primary dendrites in mitral cells during the developmental process. Dendrite pruning process fine tunes the excitatory synaptic inputs to each mitral cell, from multiple glomeruli to just one. Adopted from Figure 1B of Fujimoto et al., *Dev Cell* 2023.

Reviewer #3 (Remarks to the Author):

The methods proposed on this manuscript aims on solving two important constrains on neuronal reconstruction: (1) automation and (2) dense reconstructions within the same tissue. As an approach they combine stochastic labeling of neurons with 7 fluorescent markers and a computational strategy to cluster and identify the labeling in different neurons. On strength of this approach is that the physical continuity of the neurite fragments is less important as the combination of fluorescent proteins serves as a barcode to support the reconstruction.

The idea is very interesting, and the authors test its feasibility on dendrites and on the inter-areal portion of axons, which are usually thicker. I was surprised for not seeing a figure dedicated to the reconstruction of the thinner and more local arbors of the axons, and I think that would increase the impact and adoption of this approach.

My major recommendation is for the authors to either add local arbor axonal reconstructions to the neurons showed in Figures 6 and 7, or discuss what are limitations of the approach that at this point prevents them from obtaining the reconstructions of the fine collaterals. These fine collaterals are important because those are usually the locations of synapses.

We thank this reviewer for this important suggestion. As we used confocal microscopy, there is a trade-off between the imaging speed/area and resolution. We used 20x objective in Fig. 7 to demonstrate tracing at a millimetre scale; however, 20x falls short of imaging fine axonal fibres.

To demonstrate the reconstruction of fine axonal collaterals, we imaged a smaller area of the piriform cortex using a 63x objective lens. Fluorescence signals in axonal collaterals are often fragmented; nonetheless, axon signals from different neurons were successfully reconstructed using the QDyeFinder pipeline, as shown in Fig. R7 (also in Supplementary Fig. 11) (Lines 316-322).

We can analyse images at the synaptic scales. However, it is time-consuming to image a large area at the synaptic resolution. For example, it is practically impossible to image the entire area of the piriform cortex using 63x. This point is now mentioned in the Discussion (Line 361-363).

Fig. R7. Reconstruction of fine axonal collaterals (Supplementary Fig. 11).

Minor points

1. The authors present estimations of percentage of neurons with unique labels for different total numbers of labeled neurons. I would suggest they discuss where do they

see the sweet spot for this approach in terms of number of labeled neurons versus quality of reconstructions. When does multiple neurons labeled with the same combination of proteins becomes problematic given the quality metrics for soma, dendrite and axonal reconstructions?

We agree, and this point is now discussed in Line 395-402.

2. Related with the previous point the authors mention: “As many of neurons are uniquely labelled by the combinatorial expression of 7 XFPs, we considered that we should be able to identify neurites for different neurons solely based on the colour hue information”. How many times do the automatic reconstructions lead to reconstructions with 2 somas? This could potentially be a metric to predict the percentage of “non-unique labels” that would be tolerated.

Prediction based on synthetic data is shown in Fig. 1.

We did not use the number of soma to evaluate this point, but the numbers of clusters with multiple neurons (spatially separated) are indicated in Fig. 6c (dendrites) and Supplementary Fig. 8c (axons).

3. Page 6:” We, therefore, decided to extract colour vector data for regions of interest (ROIs) consisting of multiple pixels”. How many pixels is the minimum?

The cut-off value was determined based on the length of the neurite fragments rather than the number of pixels. This is described in Text (Line 198-200 and 370-373), Fig. 3c, and Supplementary Fig. 7b.

4. The authors could discuss or compare the size of their reconstructions with the state of the art reconstructions of the same cell types. E.g. State of the art reconstructions of cell type “X” (citation) achieve “Y” length of axon and “Z” branch points, our algorithm finds “A” length and “B” branchpoints per neuron of cell type “X” and in addition can reconstruct a total of “C” neurons within the same sample.

The size of the reconstruction will be determined by the accuracy and speed of reconstruction. However, it is very difficult to evaluate this point quantitatively. Instead, we tested several state-of-the-art methods, and the comparison is summarized in Table R1 (also in Supplementary Table 1). We believe that this is sufficiently helpful to the

audience.

Tracing strategy	Manual tracing based on physical continuity	Auto-tracing based on physical continuity	Multicolour (Brainbow/Tetbow)
Examples of software	Aivia, Amira, HortaCloud, NeuTube, NeuroLucida, Simple Neurite Tracer, Vaa3D	Arivis Vision4D, Imaris Filament Tracer, NeuroLucida 360, Vaa3D	QDyeFinder, Sümbül et al. ²⁸
Prior parameter optimization	Not required (but requires training of human skills)	Required (parameter optimization or training dataset for deep learning)	Required (clustering parameters are needed, e.g., Th(d))
Speed	Slow	High	High
Accuracy for long-neurites	High (gradually decrease)	Low (gradually decrease)	High (as long as the colour hue is consistent)
Accuracy for densely labelled circuits	High (as long as neurites are visually discernible)	Low	High (but limited to ~100 neurons when 7 XFPs are used)
Tracing broken neurites / across images	No	No	Yes

Table R1. Comparison of different tracing software for neurite reconstruction based on light microscope images (Supplementary Table 1).

5. The first author of reference 4 is Microns Consortium and not Bae.

We have corrected the author information.

Reviewers' Comments:

Reviewer #1:

Remarks to the Author:

I have reviewed the revised manuscript and the responses provided by the authors. I acknowledge and appreciate the efforts made by the authors to address the concerns raised, including the introduction of new noise analysis (Supplementary Fig. 6). While there have been efforts to address certain aspects, some of my primary concerns remain unaddressed. The evidence presented in the manuscript and supplementary materials does not fully convince me of the significant advancement of QDyeFinder and dCrawler over existing technologies. Below, I detail my specific concerns:

- I recognize the challenge in directly comparing reconstruction pipelines, especially when previous methods are not publicly available or compatible with new datasets. However, the level of comparison with other methods in the manuscript still falls short of what is typically expected in the field. For reference, the comparisons in Figs. 5 and 6 of [1] serve as good examples of the depth and rigor needed. The provided Supplementary Fig. 12, which contrasts QDyeFinder and Imaris, is limited to qualitative assessment and lacks clarity in demonstrating the superiority of QDyeFinder. The comparison focuses on a very small, zoomed-in area, raising questions about its fairness and representativeness. Furthermore, it could be argued that Imaris recovers more connections, while QDyeFinder reconstructs a smaller portion of neurites. Since detection and segmentation algorithms inherently involve a trade-off between precision and recall, a quantitative comparison such as Area Under the Receiver Operating Characteristic (AUROC), similar to what is done in Supplementary Fig. 13 for clustering algorithms, is essential for a fair evaluation. The current comparison in Supplementary Fig. 12 does not sufficiently demonstrate the effectiveness of QDyeFinder.

- I understand that practical applications often require heuristic steps. However, the key considerations are the intellectual novelty of the computational pipeline and its utility to the broader research community, not just the authors. In terms of intellectual novelty, the approaches described by the authors seem to have been previously suggested in the literature [2]. Regarding practical utility, a pipeline predominantly comprising heuristics tailored to specific needs may not be widely applicable or beneficial to the community.

- The precision-and-recall analysis presented in Supplementary Fig. 13 leaves ambiguity regarding whether dCrawler offers any meaningful advancement over existing algorithms. This is compounded by the authors' own assessment that the performance of the algorithms is broadly similar.

[1] Li et al, Precise segmentation of densely interweaving neuron clusters using G-Cut, Nature Communications (2019)

[2] Sümbül et al., Automated scalable segmentation of neurons from multispectral images, NeurIPS (2016)

Reviewer #2:

Remarks to the Author:

The authors have satisfactorily addressed my comments.

Reviewer #3:

Remarks to the Author:

The authors have effectively addressed the majority of my comments. However, I believe that replacing qualitative comparisons with quantitative ones when evaluating their methods would greatly benefit their manuscript and the broader research community. While I understand the authors' point that quantitative results can vary in terms of accuracy, speed, and other factors, the

essence of establishing a standard process is precisely to standardize these measurements. Without this standardization, qualitative assessments it is also difficult.

Summary of Revision

We thank all the reviewers for careful evaluation of our revision. To respond to the remaining concerns, we have updated Supplementary Fig. 12 and added Supplementary Fig. 14.

Supplementary Fig. 12. Comparison with existing auto-tracing software.

Supplementary Fig. 13. Comparison between DBSCAN and dCrawler.

In the rebuttal letter, the reviewers' comments are *italicised*, and our responses are provided for each comment. New changes in this revision are highlighted in both the manuscript and the rebuttal letter.

We sincerely hope that this revision is satisfactory to all the reviewers.

Reviewer #1 (Remarks to the Author):

I have reviewed the revised manuscript and the responses provided by the authors. I acknowledge and appreciate the efforts made by the authors to address the concerns raised, including the introduction of new noise analysis (Supplementary Fig. 6).

While there have been efforts to address certain aspects, some of my primary concerns remain unaddressed. The evidence presented in the manuscript and supplementary materials does not fully convince me of the significant advancement of QDyeFinder and dCrawler over existing technologies. Below, I detail my specific concerns:

- I recognize the challenge in directly comparing reconstruction pipelines, especially when previous methods are not publicly available or compatible with new datasets. However, the level of comparison with other methods in the manuscript still falls short of what is typically expected in the field. For reference, the comparisons in Figs. 5 and 6 of [1] serve as good examples of the depth and rigor needed. The provided Supplementary Fig. 12, which contrasts QDyeFinder and Imaris, is limited to qualitative assessment and lacks clarity in demonstrating the superiority of QDyeFinder. The comparison focuses on a very small, zoomed-in area, raising questions about its fairness and representativeness. Furthermore, it could be argued that Imaris recovers more connections, while QDyeFinder reconstructs a smaller portion of neurites. Since detection and segmentation algorithms inherently involve a trade-off between precision and recall, a quantitative comparison such as Area Under the Receiver Operating Characteristic (AUROC), similar to what is done in Supplementary Fig. 13 for clustering algorithms, is essential for a fair evaluation. The current comparison in Supplementary Fig. 12 does not sufficiently demonstrate the effectiveness of QDyeFinder.

We fully understand the point raised by this reviewer. In fact, there were two issues that hampered our quantitative analysis of the traces. First, the data format used in NeuroLucida and Imaris 10 Neurite Tracer were very different and data conversion was tricky. Second, neurites are typically represented by the centroid position of the anchor points within neurites. Therefore, two independent traces rarely overlap at the pixel level.

To overcome the first issue, we obtained a commercialized Imaris XTension plugin and wrote a custom code to convert the Imaris file to SWC format. As for NeuroLucida trace files, they were converted to a MATLAB struct using code included in the QDyeFinder

pipeline (**Fig. R1**).

To overcome the second issue, we intentionally increased the radius of the traces from test data (QDyeFinder and Imaris Filament Tracer) by 10 pixels. As for QDyeFinder, we used the same traces as the ground truth as detailed in Fig. 5. The percentage of neurites covered by the test traces was then calculated.

Fig. R1. Quantitative analyses of traces.

Traces in Imaris 10 Filament Tracer was converted to SWC format with custom plugin. Then, after the enlargement of the radius by +10 pixels, binary images were prepared. Neurite tracing was performed from cell bodies, as was done for ground truth in NeuroLucida. Imaris 10.1 was used. As for traces in NeuroLucida, the XML files were converted to MATLAB. After enlargement of the radius by +10 pixels (for QDyeFinder), binary images were prepared.

Fig. R2. Comparison with existing auto-tracing software

a, Automated neuron reconstruction with Imaris 10 Filament tracer and QDyeFinder. The manual tracing was done in NeuroLucida 360 as described in Fig. 5.

b, Quantitative comparison between QDyeFinder and Imaris 10 Filament tracer. Percent reconstructed indicates the area of intersection out of the area of the ground truth. Box plots indicate median \pm IQR (20 neurons). As we could only reconstruct 20 neurons in Imaris, we compared the 20 neurons. $p^{***} < 0.001$ (Mann-Whitney U-test)

c, Successfully reconstructed neurites (indicated in b) are shown.

d, Some examples of single neuron reconstructions.

Fig. R2 (also in new Supplementary Fig. 12) shows the quantitative comparison of Imaris 10 Filament Tracer vs. QDyeFinder. Here, the ground truth is the manual tracing data; while there may be more neurites from the same neuron in this image, and QDyeFinder can detect such neurites, we cannot confirm whether they are really from the same neuron. Therefore, it is difficult to evaluate false positives. We can only evaluate false negatives.

Here we examined how much of the ground truth data is contained in the Imaris vs. QDyeFinder dataset. Fig. R2b and c. We found that QDyeFinder detected significantly more neurites than Imaris 10 Filament Tracer. This is most likely because Imaris tends to connect wrong fragments when neurons are densely labelled. QDyeFinder does not make such mistakes because it relies on colour hues, but not the physical continuity of neurites.

- I understand that practical applications often require heuristic steps. However, the key considerations are the intellectual novelty of the computational pipeline and its utility to the broader research community, not just the authors. In terms of intellectual novelty, the approaches described by the authors seem to have been previously suggested in the literature [2]. Regarding practical utility, a pipeline predominantly comprising heuristics tailored to specific needs may not be widely applicable or beneficial to the community.

Intellectual novelty

As we detailed in the previous rebuttal letter, the major differences are 1) the use of 7 XFP instead of 3 XFP in Sümbül et al., and 2) the use of dCrawler instead of K-means clustering in Sümbül et al. The combination of these two parts has dramatically improved the discriminability of neurons and achieved the identification of “a single neuron” from densely labelled (~100) neurons, which has never been achieved in the previous method (Sümbül et al.). We, therefore, believe that our strategy, although based on multiple minor improvements, has achieved a major step forward in terms of performance. We understand if this reviewer does not fully agree; we know this is also science.

Practical utility

We attempted to reproduce the main segmentation algorithm described in the Sümbül et al. paper for comparison purposes. However, due to the lack of open source code, we had to develop our own implementation. Despite our best efforts, our implementation did not achieve the desired level of performance. The lack of detailed information regarding the intricate steps, parameter settings, and algorithmic choices made during the development of the segmentation pipeline posed a significant challenge. The lack of comprehensive documentation made the reproduction process arduous and time-consuming, hindering our ability to accurately replicate the results presented in the original study. We, therefore, decided to compare only dCrawler vs. k-means clustering in our QDyeFinder pipeline (Supplementary Fig. 12).

We have made all the code developed in this study publicly available (<https://github.com/mleiwe/QDyeFinder>) in the hope that this will be helpful to the community. Indeed, there are several heuristics in our pipeline (Fig. 3); however, we have confirmed that a variety of microscopes and images can be used for the default setting of our QDyeFinder pipeline. First, we have shown that linear unmixing, which is critical for the 7-color imaging, is possible for any kinds of confocal microscopes (not just Leica or Zeiss spectral imaging systems) as detailed in the previous rebuttal letter. Of course, the code for the linear unmixing is available in Github. Second, we have confirmed that $Th(d)$ of ~ 0.2 in dCrawler works well for a variety of image data with different objective lenses (20x and 63x), neurites (dendrites and axons), and labeling methods (in utero electroporation and AAV labeling). Therefore, our pipeline is not as tricky as this reviewer worries. As with other methods papers from our lab, practical details will be described in SeeDB Resources (<https://sites.google.com/site/seedbresources/>).

- The precision-and-recall analysis presented in Supplementary Fig. 13 leaves ambiguity regarding whether dCrawler offers any meaningful advancement over existing algorithms. This is compounded by the authors' own assessment that the performance of the algorithms is broadly similar.

[1] Li et al, Precise segmentation of densely interweaving neuron clusters using G-Cut, Nature Communications (2019)

[2] Sümbül et al., Automated scalable segmentation of neurons from multispectral images, NeurIPS (2016)

In Supplementary Fig. 13, we demonstrated that dCrawler performs much better than the k-means clustering used in Sümbül et al. On the other hand, we agree that the example data used in the current Supplementary Fig. 13 does not clearly demonstrate the utility of dCrawler over DBSCAN.

Here we illustrate the conceptual difference as well as performance of the dCrawler vs. DBSCAN using synthetic data in Fig. R3 (also in Supplementary Fig. 14). As was already explained in Table 1, the clear advantage of the dCrawler is that it is “threshold-based” and different clusters do not merge as long as they are far apart. This is a clear advantage,

because we assume that each neuron is labelled with a unique and consistent colour hue. For example, colour hues in neurons are not distributed as shown in **Fig. R3b, left**. We believe that dCrawler can better clusters colour hue data in neurons.

Fig. R3. Comparison of clustering algorithms in QDyeFinder.

Comparison between DBSCAN and dCrawler. Both DBSCAN and dCrawler are optimised for different situations.

a, A situation where dCrawler outperforms DBSCAN. When two clusters are close or adjacent, DBSCAN will incorporate them into a single cluster by its nature (left panel). DBSCAN only

works well when the boundaries between clusters are clear. Since dCrawler's threshold is tied to the putative centroid, the two clusters remain separate.

b, A situation where DBSCAN outperforms dCrawler. In this case, the bottom cluster (the mouth) isn't uniform in its distribution but is successfully grouped into 1 cluster by DBSCAN. dCrawler identifies 3 separate clusters instead of 1 in reality.

c, Real-world example of clustering by colours using the MATLAB example image peppers.png. DBSCAN is unable to identify individual colours as the points are spread in a spectral manner. While dCrawler is able to handle this scenario and identify 9 clusters/colours.

Reviewer #2 (Remarks to the Author):

The authors have satisfactorily addressed my comments.

We are pleased to find this comment.

Reviewer #3 (Remarks to the Author):

The authors have effectively addressed the majority of my comments. However, I believe that replacing qualitative comparisons with quantitative ones when evaluating their methods would greatly benefit their manuscript and the broader research community. While I understand the authors' point that quantitative results can vary in terms of accuracy, speed, and other factors, the essence of establishing a standard process is precisely to standardize these measurements. Without this standardization, qualitative assessments it is also difficult.

We thank the reviewer for his/her constructive comments. We totally agree. We have now performed a quantitative evaluation of Imaris 10 Filament Tracer vs. QDyeFinder. See our responses to Reviewer #1 and **Fig. R2** (also in **new Supplementary Fig. 12**). We believe that this revision is satisfactory for this reviewer.

Reviewers' Comments:

Reviewer #1:

Remarks to the Author:

I have examined the revised manuscript and the authors' responses, and I appreciate the efforts made to address my concerns. I believe that these concerns have been effectively resolved in the revised version.

Reviewer #3:

Remarks to the Author:

The authors have addressed my comments